# Rhesus macaques self-curing from a schistosome infection can display complete immunity to challenge

Murilo Sena Amaral [1], Daisy Woellner Santos[1,2], Adriana S. A. Pereira[1,2], Ana Carolina Tahira[1],
João V. M. Malvezzi [1], Patrícia Aoki Miyasato[1], Rafaela de Paula Freitas [1], Jorge Kalil[3], Elisa M. Tjon Kon Fat[4],
Claudia J. de Dood[4], Paul L. A. M. Corstjens [4], Govert J. van Dam [5], Eliana Nakano[1],
Simone de Oliveira Castro[6], Vânia Gomes de Moura Mattaraia[6], Ronaldo de Carvalho Augusto [7,8],
Christoph Grunau [8], R. Alan Wilson[9] & Sergio Verjovski-Almeida [1,2✉]

The rhesus macaque provides a unique model of acquired immunity against schistosomes, which afflict >200 million people worldwide. By monitoring bloodstream levels of parasite-gut-derived antigen, we show that from week 10 onwards an established infection with *Schistosoma mansoni* is cleared in an exponential manner, eliciting resistance to reinfection. Secondary challenge at week 42 demonstrates that protection is strong in all animals and complete in some. Antibody profiles suggest that antigens mediating protection are the released products of developing schistosomula. In culture they are killed by addition of rhesus plasma, collected from week 8 post-infection onwards, and even more efficiently with post-challenge plasma. Furthermore, cultured schistosomula lose chromatin activating marks at the transcription start site of genes related to worm development and show decreased expression of genes related to lysosomes and lytic vacuoles involved with autophagy. Overall, our results indicate that enhanced antibody responses against the challenge migrating larvae mediate the naturally acquired protective immunity and will inform the route to an effective vaccine.

[1] Laboratório de Parasitologia, Instituto Butantan, Sao Paulo, Brazil. [2] Departamento de Bioquímica, Instituto de Química, Universidade de Sao Paulo, Sao Paulo, Brazil. [3] Heart Institute, Faculty of Medicine, University of Sao Paulo (USP), Sao Paulo, Brazil. [4] Department of Cell and Chemical Biology, Leiden University Medical Center, Leiden, The Netherlands. [5] Department of Parasitology, Leiden University Medical Center, Leiden, The Netherlands. [6] Instituto Butantan, Biotério Central, Sao Paulo, Brazil. [7] LBMC, Laboratoire de Biologie et Modélisation de la Cellule Univ Lyon, ENS de Lyon, Université Claude Bernard Lyon 1, CNRS, UMR 5239, INSERM, U1210 Lyon, France. [8] IHPE, Univ. Perpignan Via Domitia, CNRS, IFREMER, Univ Montpellier, Perpignan, France. [9] York Biomedical Research Institute, Department of Biology, University of York, Heslington, York, United Kingdom. ✉email: verjo@iq.usp.br

A vaccine that provided protection against schistosome infection for an extended period would be a powerful weapon for control and ultimately eradication of schistosomiasis, a parasitic infection that afflicts over 200 million people worldwide[1]. However, the prolonged residence of adult worms in the host bloodstream, bathed in and feeding on immune effectors, attests to their supremely efficient evasion strategies[2]. Most vaccines attempt to replicate events after the primary exposure to a pathogen that naturally elicits solid immunity but, with chronic infections like schistosomiasis in humans, progress towards the goal has been beset with problems and pitfalls[3]. Initial vaccine experiments involved crude parasite extracts or single immunogenic proteins, generally trialled in the mouse model with limited success. More recently, schistosome transcriptomics[4], proteomics[5] and genome sequencing[6] changed the emphasis to the selection of candidates exposed on or secreted from the parasite[7]. Proteins emerging from these ventures include ShGST28, which progressed through Phase 3 trials in humans, with an inconclusive outcome[8], and three others (TSP2, Sm14 and Smp80 calpain) in Phase 1/2 trials[9,10]. The problems with vaccine development were underscored by the recent failure of 96 soluble, correctly folded proteins to elicit protection in the mouse model[11].

An alternative strategy is to elucidate the basis of protective immunity displayed by animal models[12]. Among such, the rhesus macaque is unique, in that a primary worm population matures and begins egg production but after a variable interval, the host response overcomes the infection and self-cure is achieved[13–15]. The model attained peak popularity in the 1960s, because it appeared to offer a route to a human vaccine, but was then gradually replaced by cheaper rodent models, despite their inherent limitations[16]. A problem with these rhesus macaque studies was the low number of replicates per group (sometimes a single animal[13,17]), linked to complex experimental designs asking multiple research questions, but the results offer tantalising glimpses into the nature of self-cure. To inform and accelerate the pace of vaccine development, the rhesus macaque model has recently been revisited, to establish the parameters of a primary infection with *S. mansoni*[18] and *S. japonicum*[19], using small numbers of animals.

We present here the results of a comprehensive analysis of the self-cure process in a cohort of 12 rhesus macaques. All animals are subjected to a single standardised experimental design involving primary infection, parasite establishment, gradual self-cure and a detailed analysis of the response after a secondary challenge at week 42 (Wk42), followed for a further 20 weeks (Fig. 1). We monitor the size of worm populations in the hepatic portal system over the time-course, using the level of the *Schistosoma* circulating anodic antigen (CAA)[20] regurgitated in the bloodstream, an assay not available in the historical studies of self-cure[13–15,17]. As the principal surrogate for worm burden, it has considerable advantages over faecal egg counts, with their inherent limitations[21,22]. We follow the levels of inflammatory and haematological markers in the bloodstream and probe IgG antibody levels to four antigen preparations from different parasite life cycle stages. Finally, we examine the differential capacity of plasma over the time-course to kill 3-to-5-day old schistosomula in culture, and to elicit genome-wide changes both in the epigenetic programme that regulates schistosomula gene expression and in the gene expression profile of these schistosomula in culture. A total of 15 different parameters are followed over 62 weeks, and these data provide a solid foundation on which to base future work aimed at identifying both the antigens mediating self-cure and the immunological mechanism(s) that allow the rhesus macaque to exhibit solid immunity after eliminating its worm population.

## Results

**Circulating antigen and faecal egg excretion show distinct profiles.** In an overview of the time course from infection (Wk0) up to challenge (Wk42), the CAA and eggs per gram of faeces (EPG) surrogates of worm burden exhibited clear and distinct profiles (Fig. 2a and Supplementary Data 1). The CAA level rose rapidly and regression of CAA values on time from Wk1 to Wk6 gave a y zero value at 1.19 weeks (8 days) for the start of erythrocyte feeding, after first worm arrivals in the portal system. Eggs appeared in the faeces at Wk6 and linear regression of EPG on time between Wk6 and Wk10 gave a y zero value at 5.8 weeks (40 days) for apparent female maturation. Wk10 was a pivot point with values for both parameters declining steeply thereafter, EPG preceding CAA by some weeks. We divided the time-course into the establishment (Wk0 to Wk10), self-cure (Wk10 to Wk42) and post-challenge (Wk42 to Wk62) phases to aid analysis. In addition, we stratified our subjects into Fast (Rh2, 3, 4 and 9) and Slow responder groups (Rh5, 6, 8 and 11) based on an unsupervised principal components analysis of data from all 15 acquired parameters (see Methods), and on their self-cure rate (see below).

**Egg deposition has the greatest impact during parasite establishment.** The CAA level of individual macaques, displayed on an Ln scale (Fig. 2b), emphasises the rapid initial growth tending to a plateau as the populations mature (arrival in the portal system is not synchronised). The combined mean plateau CAA values of the Fast and Slow groups for Wk8 and Wk10 were not significantly different (two-tailed unpaired $t$-test, $P > 0.05$). Rh1, with its lower cercarial exposure (see Methods), was clearly differentiated from the rest providing a useful outlier. The dataset up to Wk10 can be described by a logistic equation with a single variable $k$, the rate constant (Supplementary Fig. 1). The Fast group had the highest $k$ value, indicating that they reached their maximum earliest, the Slow group later and Rh1 last. Worm burdens predicted from peak CAA values (Fig. 2c) ranged from 209 to 450 adult worm equivalents, the means of Fast and Slow subgroups being 294 and 352 respectively. Given the 700 cercarial exposure, these values equate to a maturation level ranging from 30 to 70%. Notably, Rh1 was predicted to harbour only 68 worms, while Rh10 had the highest predicted value (487 worms) before it was withdrawn from the study.

Mean weights of Fast and Slow group animals at the start of the experiment were not significantly different (two-tailed unpaired $t$-test, $P > 0.05$), declining by 2.1 and 2.6%, respectively, out to Wk6, with a nonsignificant difference between the two slopes (two-tailed unpaired $t$-test $P > 0.05$). From Wk6 onwards the impact of egg deposition was evident from the significant, steady declines in weight ($F$-test: Slow, 0.00049; Fast, 0.0029). Slopes of the regressions were not significantly different (two-tailed unpaired $t$-test, $P > 0.05$), but weight loss was greater in the Fast (25.6%) than Slow group (11.8%) (Fig. 2d), the disparity being due to a continued decline for a full 4 weeks longer. The pattern of egg excretion (Fig. 2e) showed a similar trajectory to that for CAA, tending to a plateau by Wk10, with no significant difference between the Fast and Slow groups (two-tailed unpaired $t$-test, $P = 0.4714$ and $P = 0.3735$ at Wk8 and Wk10, respectively). Rh1 with its low projected worm burden was slower to produce detectable eggs (Fig. 2e) and showed only 6.1% weight loss at Wk10, returning to the initial weight by Wk12.

**Self-cure follows a negative exponential trajectory.** The negative exponential decline in both CAA and EPG levels from Wk10 denotes the start and form of the self-cure process (Fig. 2a). A linear regression on an Ln-transform of each CAA dataset between Wk10 and Wk42, provided a good model (Fig. 3a),

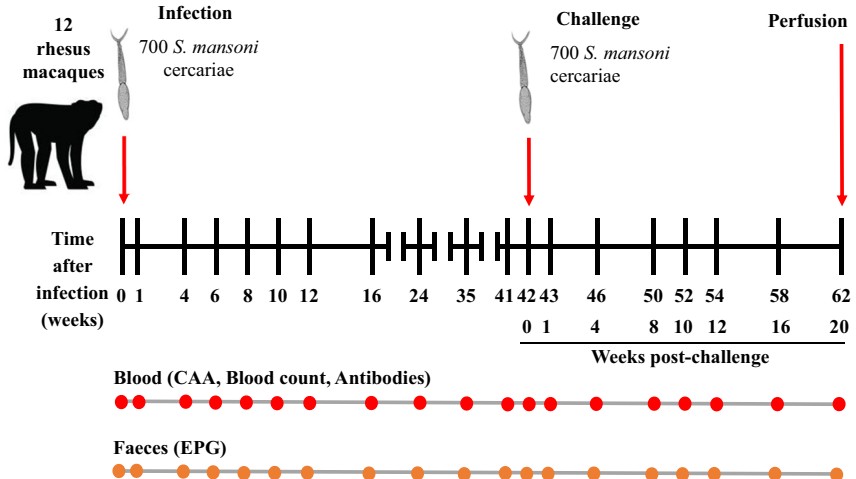

**Fig. 1 Experiment design.** Twelve rhesus macaques (*Macaca mulatta*) were each infected percutaneously with 700 cercariae of *Schistosoma mansoni* and sampled up to 42 weeks, over establishment, maturation and self-cure phases. They were then exposed to a challenge of 700 cercariae and sampled for a further 20 weeks (62 weeks post-infection). Blood was collected at the indicated time points for estimation of worm burden using the level of circulating anodic antigen (CAA), antibody titre, inflammatory markers and haematology. Faeces were also collected to determine the number of eggs per gram (EPG). At week 62 the macaques were euthanized, and the portal system was perfused to determine the actual worm burden. Macaque image is courtesy of https://www.shutterstock.com/.

allowing the cure rate to be estimated (Fig. 3b). The demise of the adult worms was a protracted process, taking a mean of 5 weeks for half to be eliminated, and so on in the manner of radioactive decay, out to Wk42. Rh1 with its lower predicted burden had a half-life of 15 weeks, while Rh6 had the most difficulty in dealing with its worm population (t½ = ~9 weeks). We stratified the rhesus subjects at the response extremes into Fast and Slow responder groups (n = 4 each; Fig. 3a, b), with Rh7 and Rh12 in an intermediate position (Rh9 was preferred to Rh12 in the Fast group because its initial worm burden at Wk10 more closely matched the other three). Regression of the mean values revealed that the t1/2 cure rate for the Fast group was ~4 weeks versus ~7 weeks for the Slow group (Fig. 3c; two-tailed *t*-test for difference of slope, $P < 0.001$).

Data for egg excretion, although less consistent, were subjected to the same analysis, noting that egg excretion had ceased in the majority of animals by Wk24. The declines in egg excretion of individual animals (Fig. 3d, e) did not parallel those of CAA (Fig. 3a). In the Fast group, Rh3 excreted so few eggs that it was omitted from analyses, whilst in the Slow group Rh6 continued egg excretion for the longest period; Rh1 had ceased excretion by Wk24 although its CAA level remained elevated. Regression of the mean Ln EPG values between Wk10 and Wk24 revealed a t½ of decline in egg excretion for the Fast group of 1.92 weeks versus 3.72 weeks for the Slow group (Fig. 3f; two-tailed *t*-test for difference of slope, $P < 0.05$). A decline in egg excretion occurred at approximately twice the rate for CAA (t½ ratios CAA/EPG, Fast = 2.1, Slow = 1.85), indicating that the worms spent some weeks in a sterile, non-reproducing state before expiring.

**Most worms fail to mature post-challenge.** At Wk42, when all macaques were challenged, the majority showed few signs of the infection, and all had regained their initial weights (Fig. 2d). The negligible detection of eggs in the faeces meant events could only be interpreted from CAA values. The low CAA levels at Wk1 post-challenge (Wk1pc), before blood feeding began, revealed that the Fast and Intermediate group responders had negligible worms surviving from the primary infection (Fig. 4a). The Slow group and Rh1 had levels of CAA above background, but still only the equivalent of a small worm population, with Rh6 the

least effective at eliminating its primary worms. By Wk4pc levels of CAA in the Fast and Intermediate groups had risen only slightly, indicating that a few worms had begun feeding on blood (a maximum of five adult worm equivalents in Rh9, seven equivalents in Rh7 and 11 in Rh12; Supplementary Fig. 2). By Wk20pc, these populations had largely been eliminated, indicating complete immunity to challenge was elicited by self-cure. The four animals in the Slow group showed higher rises in CAA level (Fig. 4a), peaking at Wk4pc in Rh5, 8 and 11 (24 to 32 adult worm equivalents), and at Wk8pc in Rh6 (49 adult worm equivalents). Rh1 showed the most rapid rise in CAA level (Fig. 4a) by Wk4pc (59 worm equivalents) and an equally rapid fall by Wk8pc with levels declining thereafter towards the background. The mean CAA level from the Slow group was significantly higher than that of the Fast group at all time points post-challenge (two-tailed unpaired *t*-test).

Few worms (nine males and two females, Supplementary Data 1) were recovered by portal perfusion of the macaques at Wk20pc. All were small and pallid, with no hemozoin pigment in the gut, so had not been ingesting erythrocytes. Compared with adult worms recovered from mice, the worms recovered from rhesus were appreciably smaller (Supplementary Fig. 3). Confocal microscopy on all recovered worms, compared with mouse worms, indicated an absence of sexual activity or reproductive capability in both sexes. Male testes were reduced in size and although spermatozoa were present in the seminal vesicles they were congealed into a solid mass (Fig. 4b.1 versus 4b.2). Ovaries and vitellaria in females were also reduced, but most telling, no egg was being formed in the ootype (Fig. 4b.3 versus 4b.4). We conclude that any surviving females in self-cured animals could not produce eggs, and hence generate hepatic pathology.

**Probing antibody responses distinguishes egg deposition from worm death.** We interrogated the profile of specific IgG production over the 62-week time course of the experiment, using four soluble antigen preparations (Fig. 5). We divided the profiles by life cycle events: skin penetration (Day 0, cercariae, SCAP); migration to, and maturation in the portal system (Wk0–Wk5, schistosomula, SSP); egg deposition (Wk5–Wk10, eggs, SEA); worm death and clearance (Wk10–Wk42, adult worms, SWAP).

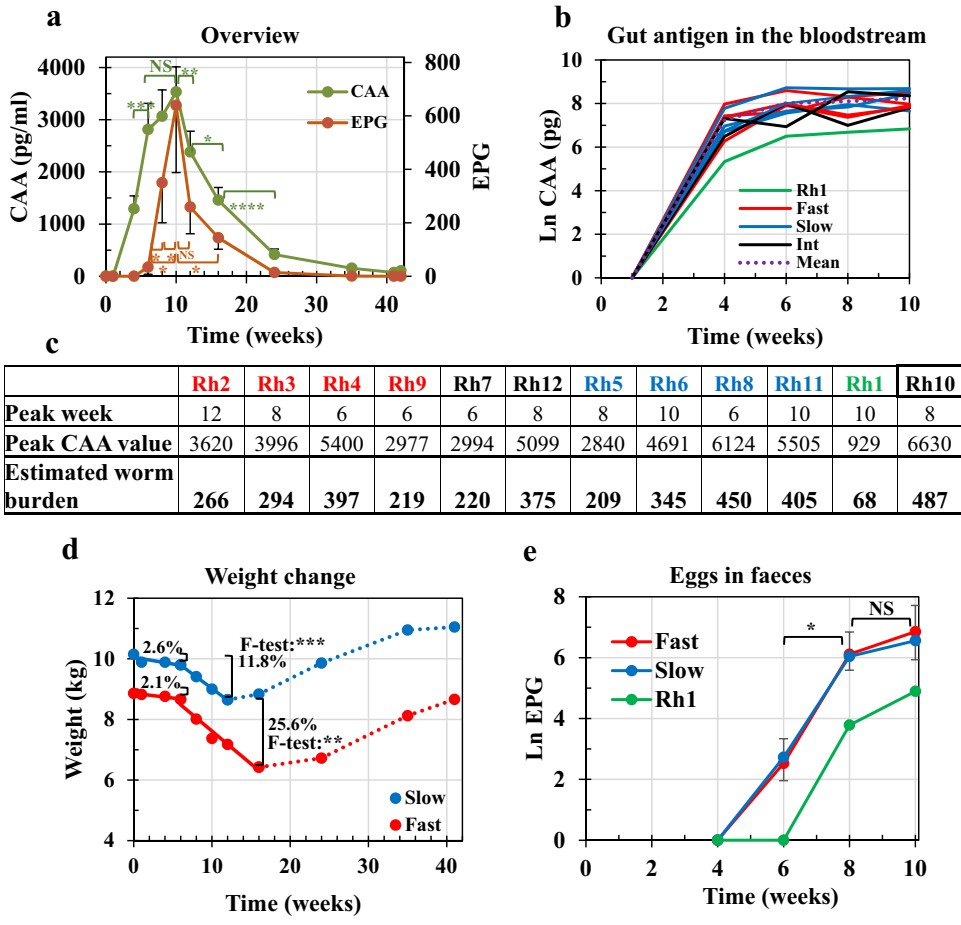

| | Rh2 | Rh3 | Rh4 | Rh9 | Rh7 | Rh12 | Rh5 | Rh6 | Rh8 | Rh11 | Rh1 | Rh10 |
|---|---|---|---|---|---|---|---|---|---|---|---|---|
| Peak week | 12 | 8 | 6 | 6 | 6 | 8 | 8 | 10 | 6 | 10 | 10 | 8 |
| Peak CAA value | 3620 | 3996 | 5400 | 2977 | 2994 | 5099 | 2840 | 4691 | 6124 | 5505 | 929 | 6630 |
| Estimated worm burden | 266 | 294 | 397 | 219 | 220 | 375 | 209 | 345 | 450 | 405 | 68 | 487 |

**Fig. 2 Overview of the primary infection and analysis of the establishment phase. a** The profile of circulating anodic antigen (CAA) and faecal egg output (eggs per gram, EPG) over the primary time course. Values are means ± SE. Statistical comparisons made using one-sided Student's *t*-test are indicated with brackets. For CAA values, $P = 0.0009$ (Wk6 versus Wk4), $P = 0.0082$ (Wk12 vs Wk10), $P = 0.022$ (Wk16 versus Wk12) and $P < 0.0001$ (Wk24 versus Wk16). For EPG values, $P = 0.0203$ (Wk8 versus Wk6), $P = 0.0219$ (Wk10 vs Wk8), $P = 0.0109$ (Wk10 versus Wk6) and $P = 0.024$ (Wk16 versus Wk10). Significance $P < 0.05$ *, $<0.01$ **, $<0.001$ ***, $<0.0001$ ****; $n = 11$ rhesus macaques, except Rh10. **b** The level of CAA in the bloodstream as an indicator of worm maturation in each animal over the establishment phase. The colours of plot lines denote stratification into Fast (red), intermediate (black) and Slow (blue) responder groups, plus Rh1 (green) based on cure rate; dotted line (······) indicates the mean of all animals. **c** The peak value attained by the CAA level in each animal, its week post-infection and the predicted adult worm burden, estimated from ref. [43]. Rh10, with the highest values, was withdrawn from the study. **d** Mean weight change of Fast and Slow groups over the primary time course. The significance of the linear regressions was determined using an *F*-test with values over Weeks 0–6 not significant; Slow, weeks 6–12, $P = 0.00049$ (***); Fast, weeks 6–16, $P = 0.0029$ (**); $n = 4$ macaques from the Slow and 4 from the Fast groups. The percentage weight loss over the respective periods is indicated by brackets. **e** The pattern of egg detection in the faeces over the establishment phase for Fast and Slow groups, plus Rh1. Statistical comparisons using one-sided Student's *t*-test, as above. $P = 0.0394$ and $0.0291$ (Wk8 versus Wk6) for Fast and Slow groups, respectively. $P < 0.05$ *. Values are means ± SE, $n = 4$ macaques from the Slow and 4 from the Fast groups. Source data are provided as a Source Data file.

Probing the response after primary infection revealed no obvious difference between SCAP and the other three preparations but, all four detected a small increment in antibody level between Wk1 and Wk4, statistically significant for SCAP and SEA (Fig. 5a, c).

The egg deposition phase was marked by a dramatic rise in specific IgG from Wk4 to a peak at Wk8, best exemplified by the SEA preparation (Wk4 cf Wk8; $P < 0.001$; Fig. 5c), but evident with the other three. Increment in reactivity was greatest for SEA (10.4-fold) and least for SWAP (7.7-fold). There was a positive relationship (Corr. Coeff. 0.79) between the peak reactivity against SEA and SCAP, but no similar relationship between SWAP and SCAP or SWAP and SEA (Supplementary Fig. 4). Stimulation of antibody production by eggs declined drastically after Wk8, with significant falls in the response detected by SEA at Wk10 and Wk12 (Fig. 5c). Anti-SCAP reactivity was similarly reduced by Wk12; the apparent smaller decline in reactivity detected by SWAP was not significant (Fig. 5d). The period of

worm death and clearance, illustrated by the anti-SWAP response, rose from Wk12 to a peak at Wk16, representing a further twofold increase in titre. In contrast, the responses detected by SCAP and SEA declined gradually from Wk12 out to Wk42 (Fig. 5a, c). It is notable that the peak reactivity detected by SSP in the Fast pool was only 50% that of the Slow pool (Fig. 5b) but neither pool showed the sharp decline after Wk8 detected by SCAP and SEA. Indeed, the profile of the Fast pool was reminiscent of SWAP, maintaining reactivity to SSP at Wk16 and Wk24 before gradually declining out to Wk42.

**Antibody responses after challenge indicate the level of protection induced by self-cure.** Antibody reactivity detected by the four preparations after the challenge is plotted as Δ, the change observed relative to the Wk42 values, so that anamnestic responses could be evaluated statistically against the primary responses. Inspection of Fig. 5 indicates clearly that the primary

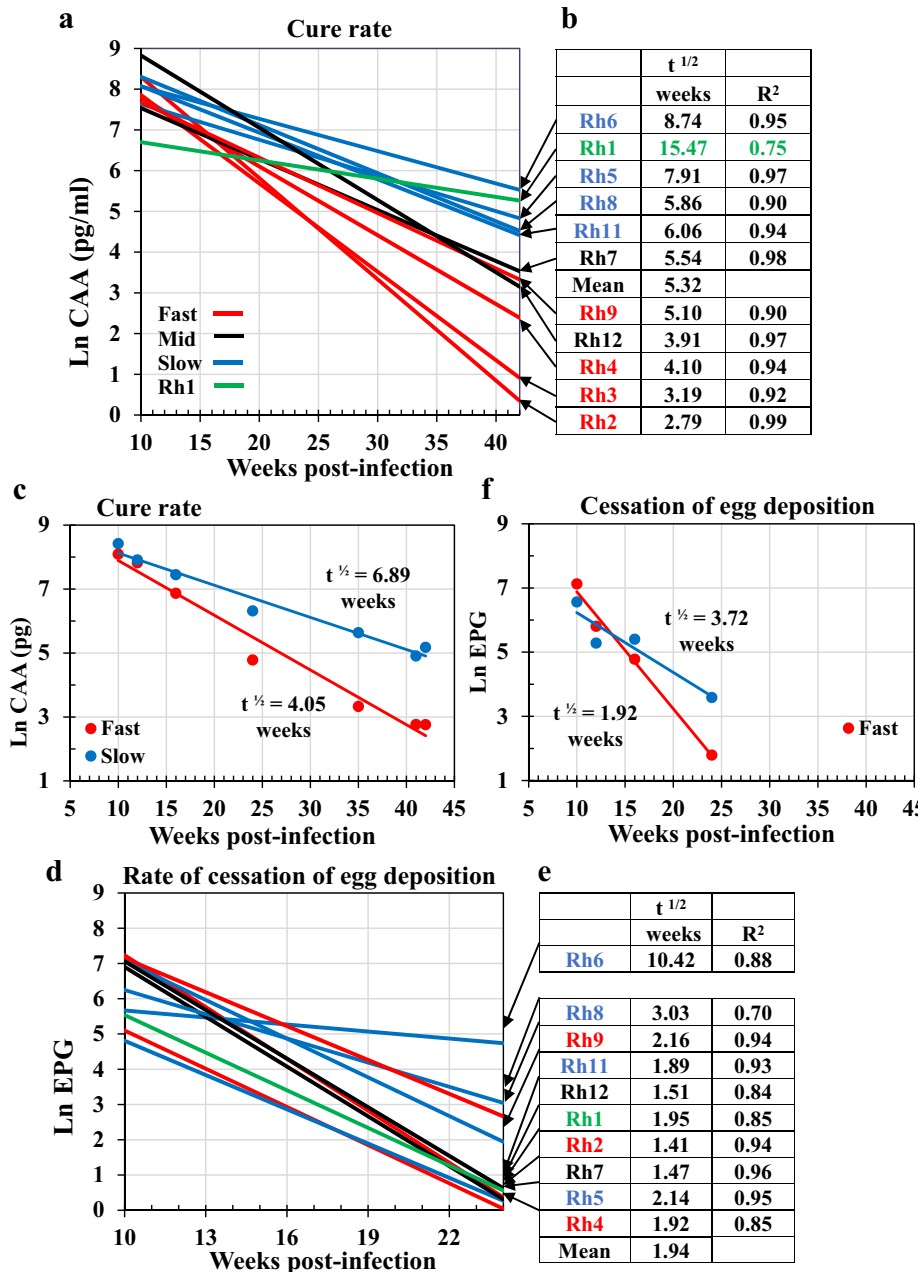

**Fig. 3 Analysis of the self-cure phase between Weeks 10 and 42 post-infection. a** Cure-rate for each animal, determined by fitting a linear regression to an Ln-transform of the negative exponential circulating anodic antigen (CAA) dataset, where t½ = ln2/λ = 0.693/regression slope. The colours of plot lines denote stratification into Fast (red), intermediate (black), and Slow (blue) responder groups, plus Rh1 (green). **b** The t½ cure-rates tabulated by value to illustrate the basis for stratification. The $R^2$ values in column three indicate the goodness of fit of each regression equation to the raw data. **c** The mean cure rate of Fast and Slow groups based on Ln CAA values. The slopes were tested for difference from each other using the Student's t-test, giving a t-value of 4.832 with 10 df and P = 0.00069; n = 4 macaques from the Slow and 4 from the Fast groups. **d** The rate of cessation of egg deposition fitted to the Ln-transform of the negative exponential eggs per gram (EPG) dataset. The plot lines are not in the same order as in the CAA stratification in **a**. **e** The t½ for the rate of cessation of egg deposition approximately ordered by value. The $R^2$ values in column three indicate the goodness of fit to each regression equation. In almost all instances the values are lower than for CAA. **f** Mean rate for cessation of egg deposition of Fast and Slow groups, based on Ln EPG values. For the cessation of egg deposition, the significance of the difference between the linear regression slopes was determined using an F-test that gave a t-value of 3.192, with 4 df and P = 0.0331; n = 4 macaques from the Slow and 4 from the Fast groups. Source data are provided as a Source Data file.

infection profiles were not reprised after the challenge at Wk42. Surprisingly, the SCAP preparation did not detect a recall response to cercarial penetration, with the value at Wk4pc significantly lower than the primary value (Fig. 5e). Similarly, the response detected by SEA at Wk4pc was not different from the primary value (Fig. 5g). In contrast, anti-SWAP responses at Wk1pc, Wk4pc and Wk8pc were significantly greater than after

the primary exposure (Fig. 5h). Additionally, the Fast pool reactivity against SSP was three times the primary value, while that of the Slow pool was only 0.5 times (Fig. 5f). The most striking differences were during the egg deposition phase detected by SEA, with the 10.4-fold increment after the primary infection replaced by a 1.4-fold increment, indicating little or no egg deposition (Fig. 5g). At Wk10pc and Wk12pc, the secondary

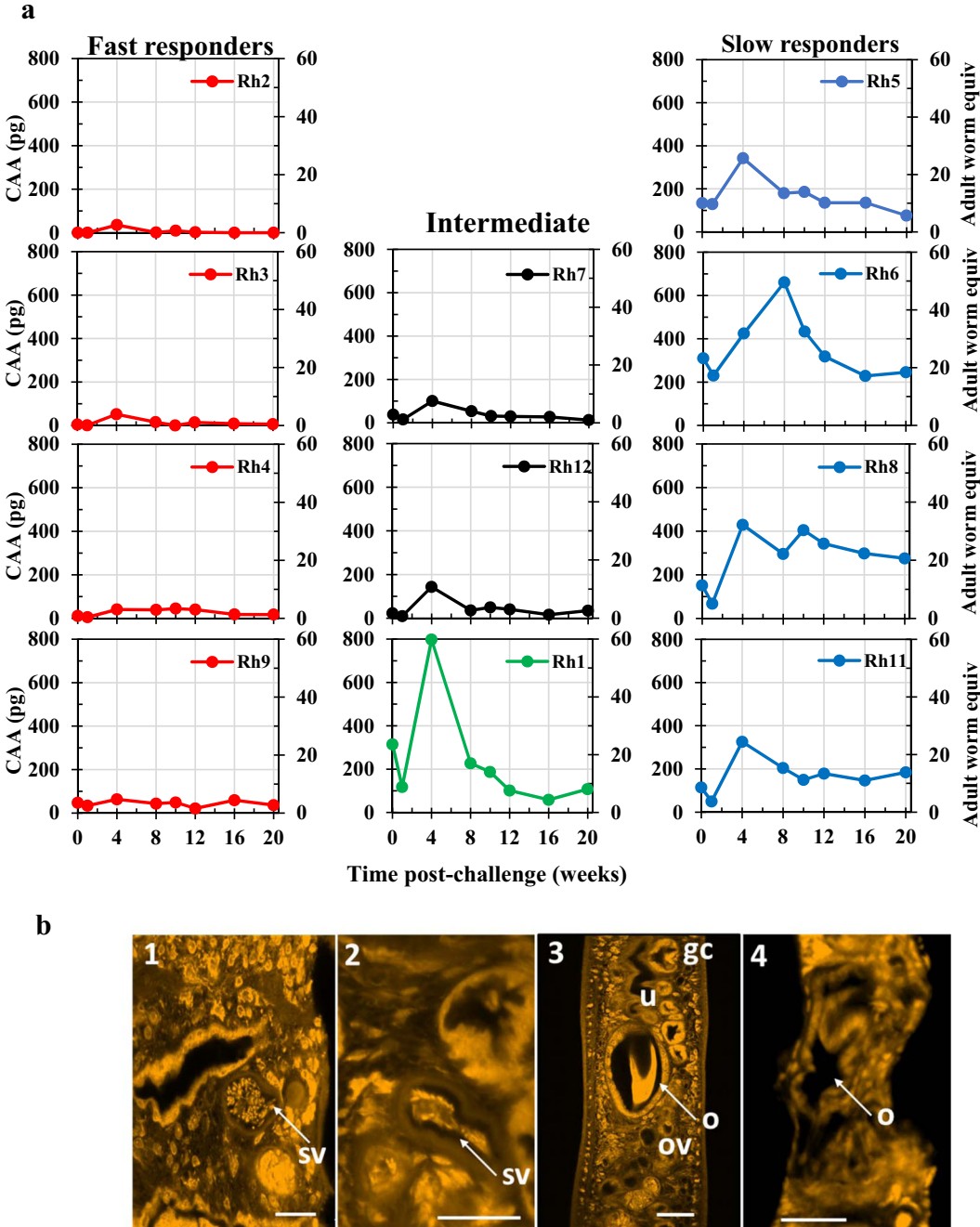

**Fig. 4 Post-challenge phase between Wk0pc (Wk42 post-infection) and Wk20pc (Wk62 post-infection). a** Estimation of challenge worm burden using circulating anodic antigen (CAA) values on the primary y-axis and predicted adult worm equivalents on the secondary y-axis, calculated from ref. [43]. On the left panels, the four individual Fast responder animals, on the right are the four Slow responders and in the centre the profile for the two intermediate responders and Rh1. Source data are provided as a Source Data file. **b** Confocal micrographs of male (1, 2) and female worms (3, 4) recovered from permissive mice (1, 3), and from rhesus macaques (2, 4) at Week 20 post-challenge. The seminal vesicle (sv) in the male worm from a mouse is full of individual stumpy spermatozoa (1) whilst that in the worm from rhesus (2) contains a compacted aggregated mass of material. The ootype (o) in the female worm from mouse (3) contains a tanned protein eggshell while the ootype in the shrunken female from rhesus (4) is empty. gc gut caecum, u uterus, ov oviduct. Magnification bars all at 20 μm indicate the shrunken size of the female worm from rhesus. Each recovered worm was prepared by staining in Langeron's Carmine, followed by dehydration, and mounting on one microscope slide. Several micrographs were obtained for each worm, and representative images are shown.

anti-SWAP response fell sharply to a level below, but not significantly different from, the primary response (Fig. 5h); by Wk16pc it was very significantly lower than after the primary infection ($P < 0.001$). The anti-SSP response detected by the Slow pool peaked at Wk8pc but then declined rapidly out to Wk20pc, while the Fast pool reached baseline by Wk16pc (Fig. 5f).

**Elevated granulocyte and lymphocyte populations accompanying self-cure were not observed after the challenge.** The haematological changes did not segregate by Fast and Slow responders, so all data were analysed as a single group. Mean haematocrit at Wk0 was 40%, declining to 35.3% by Wk12, a reduction of ~11% ($P < 0.05$; Fig. 6a). It had returned to normal

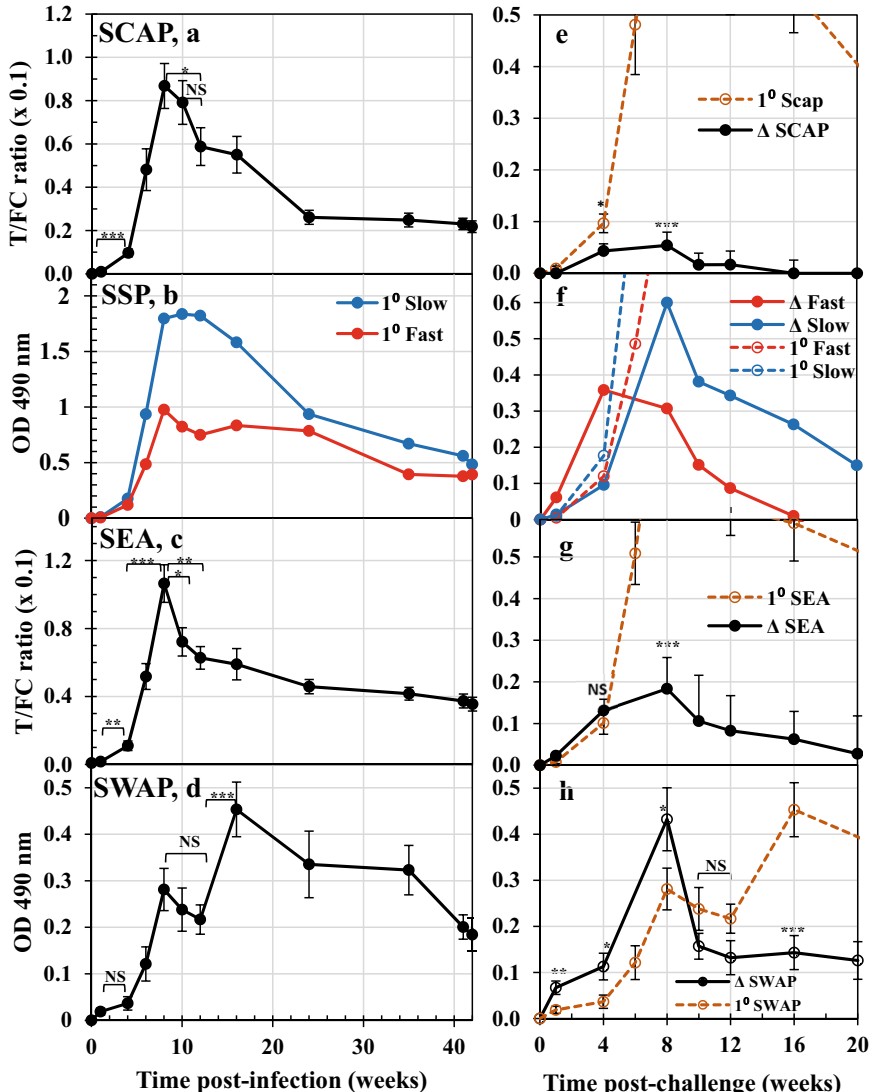

**Fig. 5 Probing antibody responses.** Responses detected over the time course of the primary infection (weeks post-infection) with soluble antigen preparations extracted from: **a** cercariae (soluble cercarial antigen preparation, SCAP); **b** cultured schistosomula (soluble schistosomula protein, SSP); **c** eggs (soluble egg antigen, SEA); **d** adult worms (soluble antigen preparation of adult schistosomes, SWAP). Values for SCAP, SEA and SWAP are means ± SE, $n = 11$ rhesus macaques, excluding Rh10. Statistical comparisons made using one-sided Student's $t$-test are indicated with brackets, and significance $P < 0.05$ *, $<0.01$**, $<0.001$***. Those for SSP are responses detected by pools of four Fast (red) and four Slow (blue) responder animals at each time point, without replication due to the low protein yield of the SSP preparation. $n = 4$ macaques from the Slow and 4 from the Fast groups. **e–h** Responses detected over the 20 weeks post-challenge using the same four antigen preparations are plotted in: **e** SCAP; **f** SSP; **g** SEA; **h** SWAP. The antibody reactivity detected is shown as the change observed (Δ), relative to the Week 42 values so that anamnestic responses can be evaluated statistically against the primary responses; data from Weeks 0 to 20 post-infection are replotted as dashed lines for comparison. Values are means ± SE, $n = 11$ rhesus macaques (excluding Rh10), except for SSP, as above. Statistical comparisons were made using a one-sided Student's $t$-test between the primary and secondary means at the selected time points indicated, with significance $P < 0.05$ *, $<0.01$**, $<0.001$***. NS not significant. Source data are provided as a Source Data file.

by Wk24 and overshot to 43.6% at Wk35 ($P < 0.05$). After the challenge, the haematocrit remained level, with a small increase to 43.3% observed at Wk8 ($P < 0.05$). Within the leucocyte populations, the most dramatic change was the marked lymphocytosis, already significant at Wk8 (Fig. 6b; $P < 0.01$), which reached a peak at Wk24 (3.54x the Day 0 value; $P < 0.001$) and remained elevated until the Wk42 challenge when it was still 1.9x higher (Fig. 6b). In marked contrast, challenge parasites elicited only a small lymphocyte response, 1.4x the challenge value at Wk8pc (Fig. 6b, $P = 0.05$, all other values NS). For monocytes, the variation between animals over time made interpretation difficult (Fig. 6b) and no statistically significant changes were observed.

Collectively, granulocytes comprised 78% of circulating leucocytes at Day 0, of which 99% were neutrophils (Fig. 6c). The absolute number of granulocytes increased to a peak at Wk10, with progressively larger proportions of eosinophils (12%) and basophils (4%) versus neutrophils (84%) ($P < 0.01$ for all three over Day 0). Eosinophils peaked (26%) at Wk24 and basophils (4.9%) at Wk12. While eosinophils, and particularly basophils, declined out to Wk42, there was a second spike of neutrophils at Wk41 (significantly higher than Wk24, $P < 0.05$). After the challenge, granulocytes peaked at Wk4pc, comprising 98% neutrophils (Fig. 6c; $P < 0.05$ over the day of the challenge). Eosinophils were never as prominent as after the primary

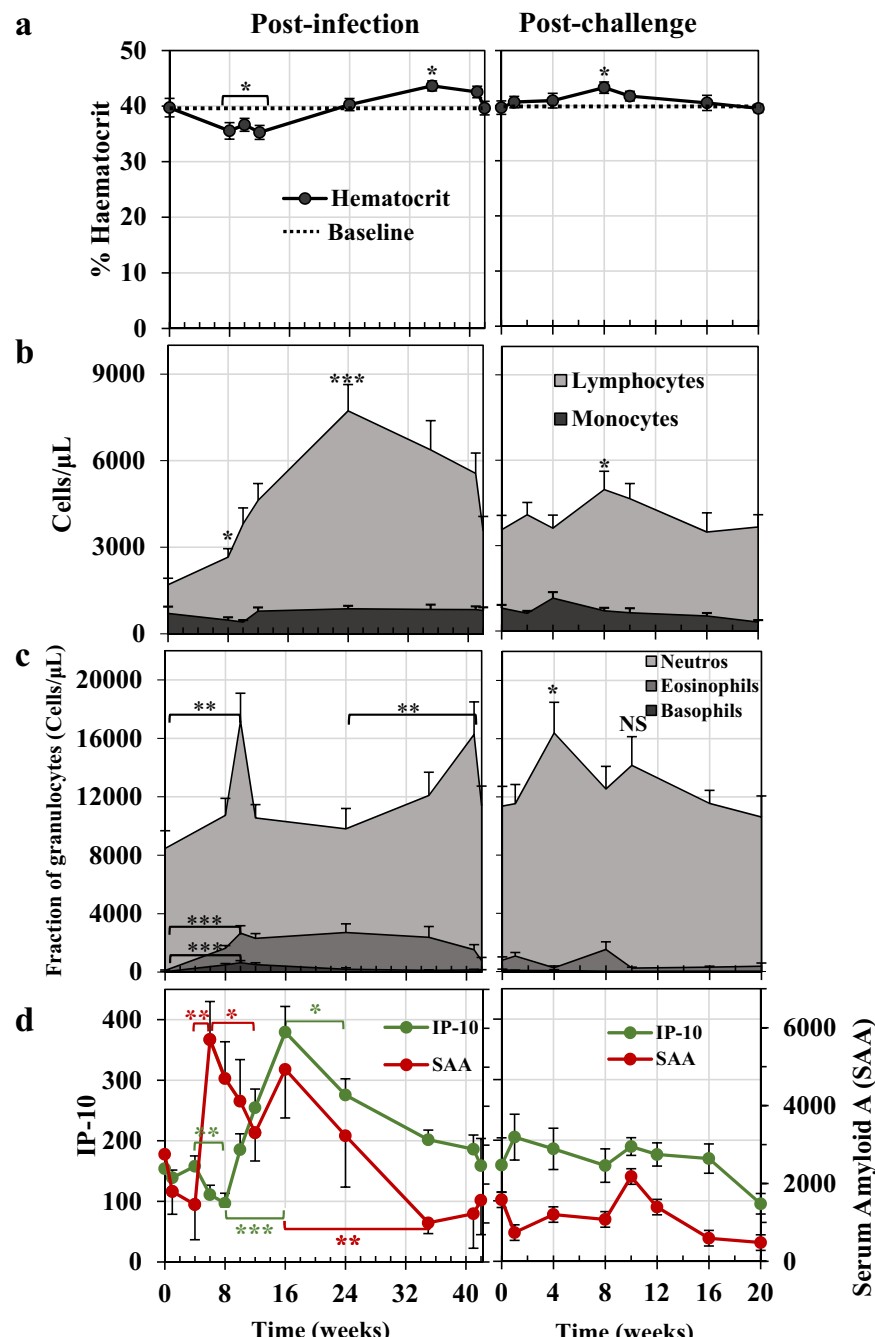

**Fig. 6 Haematology and inflammatory markers over the primary infection and secondary challenge. a** Mean haematocrit values for rhesus macaques. The baselines at Week 0 and again at Week 0 post-challenge are plotted as dotted lines to illustrate variations from the norm. Statistical comparisons using one-sided Student's t-test are with that norm, where significance $P < 0.05$ *. **b** Lymphocytes and monocytes plotted as absolute numbers per microlitre of blood. **c** Neutrophils, eosinophils and basophils plotted as proportions of the total polymorphonuclear population per microlitre of blood. Statistical comparisons using one-sided Student's t-test in **b** and **c** are with the values at Week 0 post-infection and the Week 42 challenge point, respectively, to show fluctuations from the steady-state induced by either the primary infection or the secondary challenge, respectively. Significances are $P < 0.05$ *, $<0.01$**, $<0.001$***. NS, not significant. **d** Fluctuations in the circulating level of the two markers of inflammation, interferon-gamma inducing factor 10 (IP-10/CXCL10, green line) and serum amyloid A (SAA, red line). Values are means ± SE. Statistical comparisons made using one-sided Student's t-test are indicated with brackets, and significances shown as $P < 0.05$ *, $<0.01$**, $<0.001$***; $n = 11$ rhesus macaques, excluding Rh10. Source data are provided as a Source Data file.

infection, with wide variation between animals (Wk8pc mean NS versus Wk42/Wk0pc), while basophils did not feature at all.

**The inflammatory markers SAA and IP-10 show reciprocal patterns of activity.** There was no statistically significant change

in either SAA or IP-10 over the first 4 weeks (Fig. 6d). The level of SAA then increased 3.9-fold, from Wk4 to a peak at Wk6, coincident with the onset of egg deposition ($P < 0.01$), followed by a slower decline out to Wk12 ($P < 0.05$). The apparent increase between Wk12 and Wk16 was not significant, but SAA then fell gradually to the background level by Wk32 ($P < 0.01$). After the

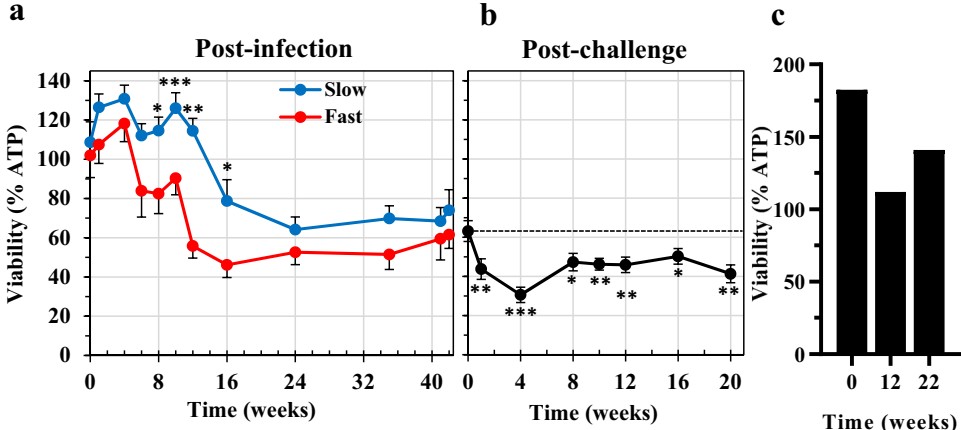

**Fig. 7 Enhanced mortality of 3-day-old schistosomula cocultured with rhesus plasma. a** The viability of 3-day-old schistosomula cultured for a further 48 h in the presence or absence of 33% v/v plasma, from Fast or Slow group animals, collected at the time points indicated over the post-infection period. Viability is expressed as % luminescence values of the test cultures, relative to the cultures without plasma. Values plotted are the means ± SE, $n = 4$ macaques from the Slow and 4 from the Fast groups; three biological replicates were assayed. Statistical comparisons between Fast and Slow groups made at each time point using one-sided Student's $t$-test; significance indicated by $P < 0.05$ *, $<0.01$**, $<0.001$***. **b** The viability of 3-day-old schistosomula cultured for a further 48 h in the presence or absence of 33% v/v plasma collected at the time points indicated over the post-challenge period. Values plotted are the means ± SE; $n = 11$ rhesus macaques, excluding Rh10; three biological replicates were assayed. Statistical comparisons of % viability were made relative to the value at the Week 42 day of the challenge, using one-sided Student's $t$-test with significances $P < 0.05$ *, $<0.01$**, $<0.001$***. **c** Control assay, showing the viability of 3-day-old schistosomula cultured for a further 48 h in the presence, relative to the absence of 33% v/v serum collected from permissive hamster animals infected with *S. mansoni*. Serum collected at time zero, and at 12- and 22-weeks post-infection; two biological replicates were assayed. Source data are provided as a Source Data file.

challenge, there was no significant upswing in the SAA level after Wk4pc. In contrast, after primary exposure, the level of IP-10 decreased significantly between Wk4 and Wk8 (Fig. 6d; $P < 0.01$), then, as the level of SAA was declining, the level of circulating IP-10 increased 3.9-fold from Wk8 to Wk16 ($P < 0.001$), indicating an approximate 2-week lag in the underlying switching mechanism. Thereafter, IP-10 levels declined out to the initial background level at Wk42 (significant by Wk24; $P < 0.05$). After the secondary challenge, as with SAA, there was no statistically significant change in circulating IP-10 levels.

**Plasma from self-curing animals causes enhanced mortality of cultured 3-day-old schistosomula**. We assessed the ability of plasma from rhesus macaques over the time course to kill 3-day-old schistosomula, using in vitro coculture assays, with total ATP level relative to untreated control cultures, as the primary readout of larval viability (Fig. 7a). Taken together, the viability of schistosomula incubated with plasma in all cultures did not decline significantly prior to Wk12 (Wk12 < Wk0, $P < 0.05$, $n = 11$). However, when Fast and Slow group cultures were analysed separately, a significantly greater killing capacity was evident in the Fast group by Wk8 ($P < 0.05$; Fig. 7a). The differential killing continued through Wk10 ($P < 0.001$), Wk12 ($P < 0.02$) and Wk16 ($P < 0.05$) before the capacity of the Slow group plasma caught up, and a plateau was reached at Wk24. After the challenge, there were negligible differences in killing capacity between Fast and Slow group plasma samples, but analysis of the total dataset revealed a further increment in killing over the level at challenge, significant by Wk1pc (Fig. 7b; $P < 0.01$). The value at Wk4pc represents a 50% reduction in schistosomula viability compared to the post-infection levels (from 63 to 31%; $P < 0.001$), and the difference from Wk0pc remained significant from Wk1pc out to Wk20pc (Fig. 7b). As a negative control for the ATP assay, 3-day-old schistosomula were incubated with serum from permissive hamsters, collected at Wk0, Wk12 and Wk22 post-infection with *S. mansoni*. These sera

had no significant impact on schistosomula viability compared with those grown in medium only (Fig. 7c).

To confirm viability reduction and killing, schistosomula cocultured in vitro with pools of rhesus plasma collected at Wk1 post-infection or Wk10pc from the Fast or Slow groups were stained with PI/FDA and observed under the microscope. Staining with PI and not with FDA confirmed the killing of plasma-treated schistosomula (Supplementary Fig. 5a). Rhesus plasma from Wk10pc killed a higher proportion of schistosomula when compared with Wk1 post-infection (Supplementary Fig. 5b).

**Plasma from self-curing animals causes epigenetic and gene expression reprogramming of cultured 3-day-old schistosomula**. Application of ChIP-Seq to 3-day-old schistosomula cocultured for 2 days with rhesus plasma collected at Wk10 showed a significant change in H3K4me3 abundance (FDR <0.05) at 76 different genomic regions when compared to Wk0 plasma (Fig. 8a, left panel; affected genes annotated in Supplementary Data 2). GO enrichment analysis (FDR <1%) identified the top ten most enriched Biological Processes (BP) as including actin cytoskeleton maintenance, foregut morphogenesis and negative regulation of autophagy (Fig. 8b left and Supplementary Data 3). An example of two genes (Smp_155780.1 and Smp_207030.1) divergently transcribed from a locus on chromosome 4 is illustrated in Fig. 8c. Both are annotated as 'GTPase-activating protein-related', having H3K4me3 marks at their transcription start sites (TSSs) (Fig. 8c, Wk0), which were lost upon exposure of schistosomula to Wk10 plasma (Fig. 8c, Wk10).

When plasma collected at Wk10 was compared with plasma from Wk8 (the earliest time point at which decrease in viability of schistosomula occurred; see Fig. 7a), a change in H3K4me3 abundance (FDR <0.05) at 116 different genomic regions was observed (Fig. 8a, right panel; annotated in Supplementary Data 4). GO enrichment analysis (FDR <1%) identified the top ten most enriched GO Biological Processes as related to genes involved with apoptotic chromosome condensation, epigenetic

**a**

**H3K4me3 chromatin mark with significant abundance changes**

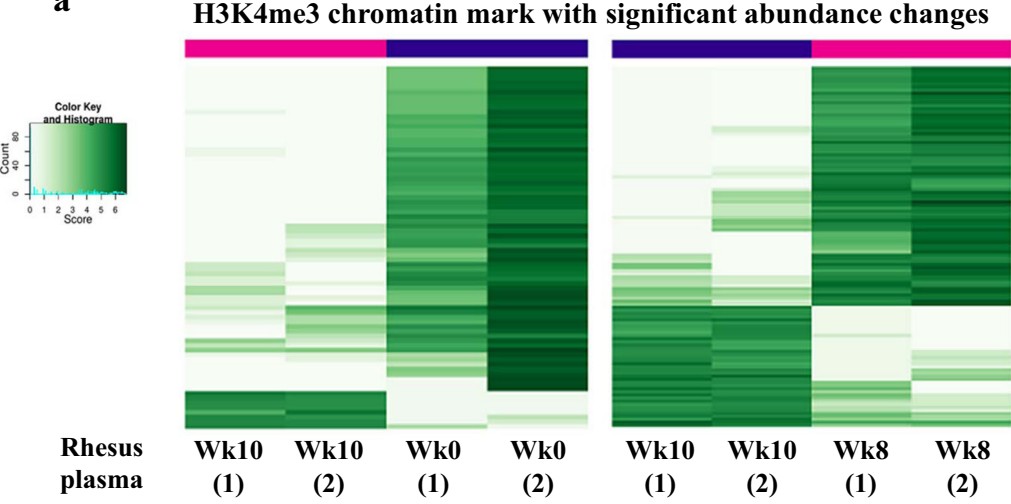

| Rhesus plasma | Wk10 (1) | Wk10 (2) | Wk0 (1) | Wk0 (2) | Wk10 (1) | Wk10 (2) | Wk8 (1) | Wk8 (2) |

**b**

**GO enriched Biological Processes of genes with significant abundance changes in H3K4me3 chromatin mark**

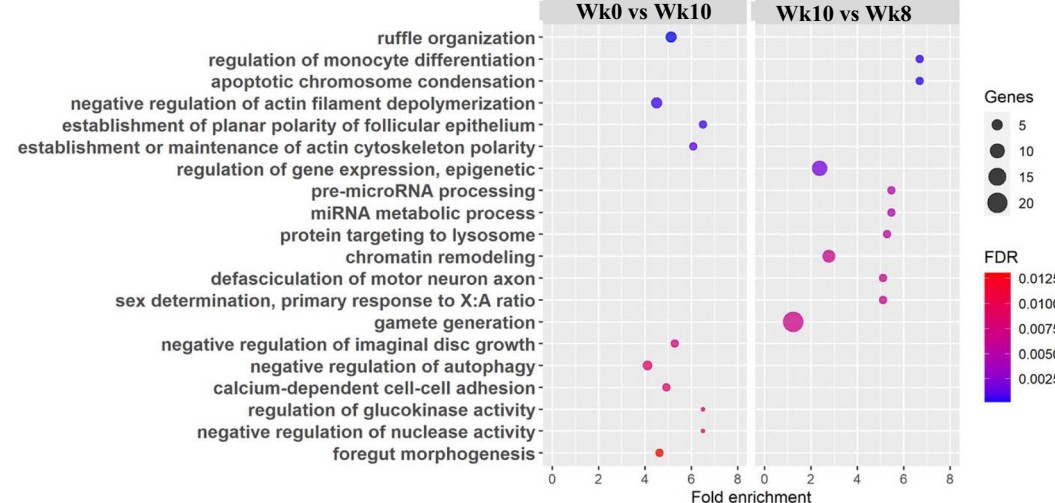

**c**

**Example of genomic loci of two genes with significant abundance changes in H3K4me3 chromatin mark**

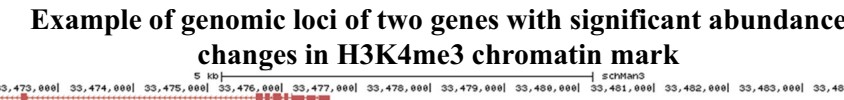

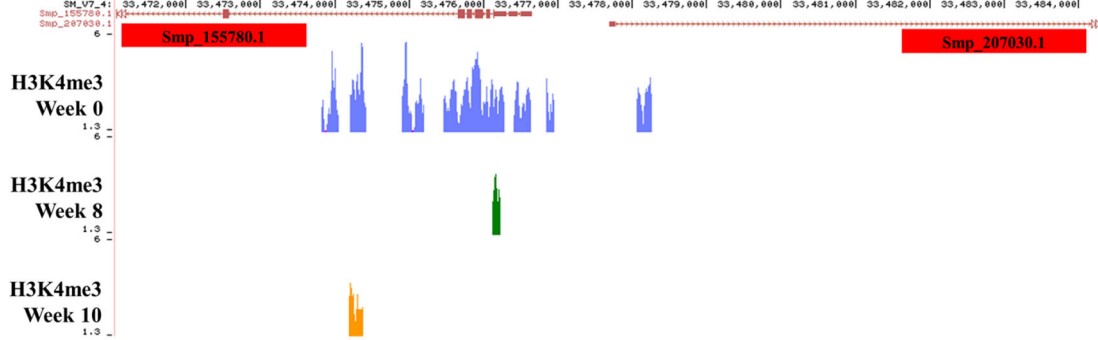

regulation of gene expression and chromatin remodelling (Fig. 8b, right and Supplementary Data 5).

To gain further insights into the mechanisms of schistosomula killing, we applied RNA-Seq to 3-day-old schistosomula cocultured in vitro with rhesus plasma collected at Wk0, Wk8 or at Wk1pc and to control schistosomula (Supplementary Data 6). A total of 79 differentially expressed genes (DEGs) were identified when schistosomula were cocultured with Wk8 plasma,

compared with control schistosomula but no enriched GO categories were found among them (Supplementary Data 7 and Supplementary Fig. 6). Fifty of these genes (63%) were also differentially expressed in schistosomula cocultured with Wk1pc plasma compared with control schistosomula, together with a further 243 DEGs (Supplementary Data 7 and Supplementary Fig. 6). Remarkably, only three GO categories were enriched among the total of 293 DEGs in the schistosomula cocultured

**Fig. 8 Changes in H3K4me3 chromatin mark occupancy at gene loci in 3-day-old schistosomula cocultured with rhesus plasma. a** The abundance of H3K4me3 histone marks at each gene locus (one gene per line) was determined by ChIP-Seq in schistosomula cultured for a further 48 h in the presence of 33% v/v plasma collected from rhesus at Week 10 post-infection, and compared with the H3K4me3 abundance in schistosomula cultured with plasma from Week 0 (left panel) or plasma from Week 8 (right panel); only the genes with significant changes in chromatin mark abundance in the comparisons are shown. Chromatin peak abundance at each gene is colour-coded; light green represents low peak counts and the dark green represents high counts (see colour key). **b** Gene Ontology classification of significantly enriched (FDR <0.0125) Biological Processes of genes with changes in H3K4me3 abundance in schistosomula cultured with rhesus plasma. Left panel, enriched GOs in the comparison between treatment with Week 10 vs Week 0 plasma; right panel, enriched GOs in the comparison between treatment with Week 10 vs Week 8 plasma. **c** Snapshot of a *S. mansoni* genome browser image (http://schistosoma.usp.br), showing a region spanning 5 kb on chromosome 4, where two protein-coding genes Smp_155780 and Smp_207030 are located (top red track). Note that these genes map head-to-head in this locus, i.e. they are transcribed in opposite directions, with their transcription start sites (TSSs) near each other in the centre of this locus region. Below the red track, three other tracks show the H3K4me3 ChIP-Seq peaks detected in the 3-day-old schistosomula cultured with plasma collected at Week 0 (blue track), Week 8 post-infection (green track) and Week 10 post-infection (orange track).

with Wk1pc plasma, all representing GO cellular components related to late autophagic processes: lytic vacuole, lysosome and vacuole (Supplementary Data 8). Among the ten genes in these three GOs, one (Smp_049150.1, Aspartate aminotransferase) was upregulated. The other nine genes were downregulated (Supplementary Data 7, yellow), including glycosylases, a phospholipase B-like and lysosome-associated membrane proteins. Manual curation identified 19 Smp genes (a total of 26 isoforms) that are orthologs of 15 genes of the early autophagy pathway but the expression level of none of them were affected by the plasma treatments (Supplementary Data 9).

## Discussion

Our ability to measure the level of gut-derived CAA regurgitated into the blood as an accurate estimate of parasite burden over the time course provided the key parameter to assess the dynamics of the self-cure process, preferable to faecal egg counts. These were previously considered a poor index of the intensity of infection[21], due to the variability of tissue egg accumulation versus egg excretion[22], the loss of appetite in some animals[21] and diarrhoea in others[23]. We estimated the inception of blood feeding on Day 8, so allowing a 48 h transition from migrating to erythrocyte-feeding form[24], the first schistosomula arrived in the portal system on Day 6. Thereafter, growth was rapid, with increasing expulsion of CAA into the bloodstream. We estimated that the first excretion of eggs occurred at 40 days, comparable to the 36.9 and 39.5 days reported previously[21]. Allowing for egg development in the tissues (5–6 days)[25], the first female worms matured at Day 34, ~26 days from the onset of blood feeding[24], so the last arrivals would be maturing around Day 54. Over this period there was minimal observable impact of the schistosome infection on the animals.

In contrast, from Wk6 onwards, the egg shedding process resulted in weight loss with bleeding into the intestinal lumen and associated dehydration. The 11% reduction in haematocrit by Wk12, agrees with the 14% recorded at 12 weeks after a 750 cercarial exposure[21]. Similar weight loss (or a lack of weight gain) was reported for *S. bovis* infections of sheep and goats[26], and calves[27], along with the elimination of some primary worms, additional retention of eggs in the tissues, and 90% suppression of egg output in surviving worms[27]. However, unlike rhesus macaques, the immune system in livestock does not deliver the decisive killer blow to *S. bovis* worms. Why the Fast group should lose more weight over a longer period than the Slow group is unclear since neither CAA level nor faecal egg production suggests a higher worm burden up to Wk10. The logistic analysis indicates that the worm biomass in the Fast group is constrained earlier than in the Slow group. The lower anti-SEA antibody response at Wk8 in the Fast group, indicating a lower fecundity of the females, appears to rule out differential tissue egg retention by

Fast group animals (cf.[21]). Their subsequent history suggests they are the ones best able to control schistosome infection, potentially by mounting a more aggressive immune attack.

The inception of self-cure around Wk10 in almost all animals suggests that antibody levels against key target antigens have reached a threshold 'immune pressure'[18] above which worm survival is affected (see below). The Fast and Slow groups represent the two extremes with ~4-week and ~7-week half-lives. That the intensity of the immune response triggered by the primary infection is the principal host determinant of the elimination process can be inferred from the lower projected adult burden and longest worm half-life (15.5 weeks) of Rh1. Similar anecdotal conclusions about the effect of the initial burden on worm longevity were reached in earlier studies[28,29]. *S. mansoni* worms are long-lived in human populations (up to 32 years)[30] with a Type I survivorship curve[31], whereas the rhesus macaque presents a rare example of a sudden switch to a Type III curve after 10 weeks, with parasite maximum longevity reduced to a few months.

The negative exponential decline in CAA level revealed that the majority of worms in a given animal succumbed rapidly to 'immune pressure', whereas a minority showed prolonged survival, pointing to genetic heterogeneity. Previous morphological analysis of surviving worms at 18 weeks showed that while many were no longer feeding on blood and apparently starving to death, others looked normal[18]. This heterogeneity could result from differential expression of a multiplicity of antigenic targets[18] and/or variations in the expression of worm-protective antioxidant proteins[32]. Our observation of chromatin structure reprogramming in schistosomula exposed to plasma from Wk8 to Wk10, may also reflect a strong immune attack on the schistosomes. Changes in H3K4me3 occupancy at a number of gene loci involved with epigenetic regulation of gene expression and chromatin remodelling suggest that these processes may be early events that lead to worm death. Two of the genes with H3K4me3 chromatin mark loss, Smp_155780.1 and Smp_207030.1, are more highly expressed in schistosomula and mature adults (>42-day-old)[33]. They both have a TBC GTPase-activating protein domain, acting on Rab-like GTPases, which play important roles in the regulation of autophagy[34], an essential process for animal homoeostasis.

There is morphological evidence for induction of autophagy in schistosomes, in response to stress conditions such as starvation or drug treatment[35–37]. In addition, some components of the autophagic machinery have recently been described in *S. mansoni*[38], but overall the process is under-researched. During the crucial period of parasite establishment investigated here post-challenge, there are two transitions that involve tissue remodelling. The removal of the spent acetabular glands during the transformation from cercarial body to skin schistosomulum

over the first 72 h is coincident with the appearance of acidic compartments representing autophagosomes[39]. Subsequently, in the pulmonary vasculature between 3 and 6 days, the schistosomulum undergoes a dramatic elongation to facilitate passage through organ capillary beds; this involves a change in circular-longitudinal muscle fibre configuration and removal of the sub-tegumentary fibrous interstitial layer[40]. Of note, Smp_090090.1, Serpin peptidase inhibitor was the protein-coding gene most highly upregulated (11.4-fold) in Day 6 schistosomula cocultured in vitro with Wk1pc rhesus plasma (Supplementary Data 7), which could impair the activity of the proteases removing the fibrous interstitial layer, so impeding migration. The expression levels of genes encoding early components of the autophagy pathway in Day 6 cultured worms were unaffected by Wk1pc rhesus plasma, whereas the levels of nine out of the ten genes involved in late stages of autophagy were downregulated. A third phase of tissue remodelling (not investigated) provides a further target when the migrating schistosomulum transitions to the blood-feeding juvenile in the portal distributaries of the liver, with the removal of the head capsule muscles and spent head gland. The structural alterations, potentially relying on autophagy, tie in with our observation of plasma-induced autophagy dysfunction in vitro suggesting that immune pressure in vivo may retard larval development and ultimately result in the observed worm starvation[18], leading to parasite death and to rhesus self-cure.

Regarding the contribution of host cellular responses, studies with other primates have shown very little detectable response of peripheral blood lymphocytes after exposure to a large cercarial dose, either by way of proliferation or cytokine production, until the onset of egg deposition (challenge controls in ref. [41]). It is the major antigenic stimulus provided by the egg, which elicits an intense inflammatory response promoting passage through the intestinal tissues to the lumen[42]. Fluctuations in the two circulating markers of inflammation, SAA and IP-10, support a concept of two different forces at play over the primary time course. The absence of changes over the first 4 weeks underscores the minimal inflammatory stimulus provided by the developing parasites, while egg deposition caused a surge in SAA level, depressing IFNγ-induced IP-10 and revealing the emerging dominance of Th2 cytokines[43]. A similar switch was noted in acute human schistosomiasis, related to parasite burden[44], and is best characterised in the murine model[45]. In rhesus macaques, the Th2 dominance is short-lived with the removal of the antigenic stimulus resulting from the 50% reduction in egg output between Wk10 and Wk12. This takes the brake off IP-10 production, and its rise up to Wk16 signals the death of half the adult worm population. The decline in the level of both markers after Wk16 parallels the resolution of the primary infection.

Circulating granulocytes could also contribute to the demise of adult worms via their consumption of ~21,000 leucocytes/day[46], potentially attacking worms from within. Indeed, a major function of the >30 proteins secreted from schistosome oesophageal glands is to disable ingested leucocytes and platelets before they reach the gut lumen[47,48]. In both rhesus macaques self-curing from *S. japonicum*[49], and in mice exposed to the radiation-attenuated cercarial vaccine[50], these secreted proteins elicit strong antibody responses that may neutralise their disabling functions, allowing intact leucocytes to reach the gut lumen. All three granulocytes can generate cytotoxic reactive oxygen species and other toxic proteins, and their high numbers at Wk10, coincident with the start of self-cure, suggests a causative link.

Antibody responses to secreted proteins may hold the key to self-cure. The consecutive life cycle stages expose the host to very different amounts of antigen (Summarised in Supplementary Fig. 7a for a 700 cercarial infection). These range from a ~6 μg shot of cercarial secretions through ~12 μg/day from developing/

blood-feeding worms to ~170 μg/day of eggs exported by females. This >14-fold discrepancy partly explains the dominance of eggs in the stimulation of antibody production. The contribution of eggs was replaced over the Wk10–Wk15 period by ~4000 μg of antigens released by adult worm death. SCAP failed to detect a unique response to primary (or secondary) cercarial exposure, suggesting that its specific constituents play little role in protection. The immune system is next primed by larval secretions[50,51] detected by the SSP preparation but the muted antibody responses up to Wk4 suggest that these secretions are not strongly immunogenic. In contrast, the dramatic rise in anti-SEA response reflected the dominance of egg-derived proteins, both in quantity and immunogenicity. However, there is little to suggest that these drive the self-cure process since mice vaccinated with live eggs are not protected[52,53]. Priming by head gland secretions ends with the start of blood feeding, but the contributions of the alimentary tract and tegument continue after Wk5, despite egg dominance, as revealed by the increasing anti-SSP values. Worm death, from Wk10 onwards in all animals, was tracked by the SWAP preparation down to Wk42. Demonstration of the cytotoxic potential of rhesus plasma, revealed by the schistosomula in vitro killing assays, was also evident in Fast responders as early as Wk8, coinciding with the first indication of adult worm stress, but the potency of Slow responder plasma lagged until Wk16. This differential matches the respective cure rates of 4 and 7 weeks between Fast and Slow groups, and points to heterogeneity in the antigens targeted.

We questioned how challenge parasites are eliminated. The very limited response of the self-cured rhesus macaques to a 700 cercarial challenge after Wk42 might at first sight seem surprising. However, it is a testament to the effectiveness of the protection elicited by the self-cure process. Rather than weight loss after challenge, there was a 5% increase, coupled with a small rise in haematocrit at Wk8pc. Both the absence of significant fluctuations in SAA and IP-10 inflammatory markers and the lack of an anti-SEA response after challenge confirm the failure of challenge females to begin oviposition and the absence of perceptible egg shedding into the gut lumen. Furthermore, there was no large biomass of dead juveniles or adults to be disposed of by inflammatory processes, such as those detected after primary exposure.

Antibody responses to the other preparations after the challenge were also muted. The rapid rise in Fast group titre detected by SSP indicates an anamnestic response, peaking at Wk4pc. A minimal commensurate rise in CAA levels suggests that proteins mediating protection are numbered among the specific SSP constituents. Early and stronger secondary response to SWAP, enriched in internal housekeeping proteins, charts the release of somatic antigens ensuing from the death of challenge worms. The increases in killing power of plasma detected as early as Wk1pc, provide crucial evidence for enhanced antibody responses against the migrating larvae. Additionally, peak killing at Wk4pc implicates pre-adults in the portal tract as targets. These conclusions are supported by a hallmark histopathological study[54], which showed that parasites arriving in the lungs of naïve rhesus macaques provoked minimal inflammation, whereas in self-cured animals they were initially detained in arterioles and capillaries by small inflammatory foci[54]. The reactions are reminiscent of those described in the lungs of mice exposed to the radiation-attenuated cercarial vaccine versus naïve controls[55,56] except that in self-cured macaques, a proportion of schistosomula continued their migration to the hepatic portal system but had failed to mature by 6 weeks. In some animals they had all been eliminated by 9 weeks[54], most likely the equivalents of our Fast responder group. This points to a major distinction between protection in vaccinated mice and self-curing rhesus macaques. Both hosts

generate inflammatory foci that prevent a proportion of larvae from migrating beyond the lungs[55,56]; only the rhesus macaque subsequently eliminates blood-feeding stages from the portal tract, potentially via antibody-mediated mechanisms. Understanding the immunological responses and molecular targets underlying the protection elicited by self-cure in the rhesus macaque must surely inform the route to an effective vaccine.

## Methods

**Ethics statement**. Housing conditions of the rhesus macaques and experimental protocols used in the study were in strict accordance with the Ethical Principles in Animal Research adopted by the Conselho Nacional de Controle de Experimentação Animal (CONCEA) and were approved by the Institutional Animal Care and Use Committee of Instituto Butantan (CEUAIB 1388/ 15). The study was carried out in compliance with the ARRIVE guidelines. The design and execution of the study complied with the recommendations of the Weatherall report (2006) The use of non-human primates in research, in which there are sections dealing with the continued need for primates in schistosomiasis research, particularly in vaccine development. The study also complied with principles set out in the UK NC3Rs Guidelines Primate accommodation, care and use (revised version, October 2017) (http://www.nc3rs.org.uk/primatesguidelines). Twelve adult female rhesus macaques (*Macaca mulatta*) from the captive-breeding colony at the Central Animal Facility at Butantan Institute were group-housed for the whole experiment, permitting social interactions, continuous socialisation and colony welfare. The facility is accredited by Conselho Nacional de Controle de Experimentação Animal (CONCEA). The animals had free access to drinking water and vitamins were provided. They had a well-planned diet, with balanced rations, including various fruits, leafy greens, vegetables, grains (sunflower, corn) and eggs. Environmental enrichment consisted of tyres and plastic barrels, suspended along the top as toys (swings), wooden coils and trunks placed on the floor, as well as fire hoses fixed in the cage ceiling. The rhesus macaques were observed daily to assess welfare using established evaluation criteria, including behaviour (body postures, facial expressions, vocalisations and interactions between animals); food and water intake; stool consistency; clinical signs of pain or discomfort.

Housing conditions of the hamsters and experimental procedures used in this study were also in strict accordance with the Ethical Principles in Animal Research adopted by the CONCEA and the experimental protocol was approved by the Ethics Committee for Animal Experimentation of Butantan Institute (CEUAIB n˚ 6748040515). Housing conditions and experimentation with mice followed the recommendations from the Biology Department Ethics Committee, University of York, and experiments were performed on personal (PIL 50/592) and project licences (PPL 60/4340) issued to RAW.

**Parasite exposure and sampling regime**. The 12 rhesus macaque females used in this study had a mean age of 13.9 ± 2.8 years and a mean weight 9.3 ± 1.9 kg at the outset. Each animal was exposed to 700 *S. mansoni* cercariae from a BH isolate (Parasitology Laboratory, Butantan Institute, Brazil) maintained by passage through golden hamsters (*Mesocricetus auratus*) and *Biomphalaria glabrata* snails. Exposure of macaques to cercariae was performed via a metal ring placed on the shaved abdominal skin, for 30 min, under ketamine hydrochloride (10 mg/kg body weight) and xylazine (0.5 mg/kg body weight) (Sespo, Sao Paulo, Brazil) anaesthesia. Of note, accidental leakage of liquid from the metal ring on Rh1 resulted in this animal receiving a lower dose of cercariae. After 30 min exposure, water was pipetted off the skin and inspected for non-penetrant cercariae. A maximum of 5.8% of non-penetrant cercariae was found in Rh4 and there was no significant correlation between non-penetrants and worm burden, estimated by the plasma level of circulating anodic antigen (CAA) at Wk10 (see below) (Spearman correlation, $r = 0.1086$; $P = 0.7366$). At Wk42, a 700 cercariae challenge was performed on all macaques, which were then followed up to perfusion at Wk62 (Wk20pc). Blood and faeces were collected at 19 different time points (Fig. 1) from Wk0 (before infection), through Weeks 1, 4, 6, 8, 10, 12, 16, 24, 35, 41, 42 after infection, and at Weeks 1 (43), 4 (46), 8 (50), 10 (52), 12 (54), 16 (58) and 20 (62) post-challenge. All the sampling, weighing and infection procedures were performed on anaesthetised animals. Blood was collected from the femoral vein of each animal in: (1) BD Vacutainer CPT tube (BD, 362753), in which plasma was separated and then stored at −80 °C; (2) BD Vacutainer EDTA (BD, 367841) for complete blood count. As animals were free-housed, faeces were collected from the rectal ampulla. The eggs are shed from the intestinal tissues into the gut lumen to be voided in the faeces, with concomitant blood and fluid loss. Mild to moderate dehydration was observed in all four Fast group animals (Rh2, 3, 4, 9), two Slow group (Rh6, 11) and one intermediate animal (Rh12). Subcutaneous rehydration therapy was required by three animals (Rh3, 6, 12) showing signs of dehydration at the peak of infection between Wk10 and Wk16. Rh10 became seriously unwell after Wk10 and was withdrawn from the study. Subsequent determination of CAA revealed that it had the highest CAA levels of all animals at Wk10.

Five golden hamsters (*M. auratus*) were infected with 100 cercariae each, to be used as permissive infection controls. Serum from hamsters was collected at Wk0, Wk12 and Wk22, and perfusion was performed at Wk22. Control reference worms

were recovered from C57BL/6 mice 7 weeks after exposure to 200 cercariae, fixed in formal saline and processed for microscopy as described in ref. [18].

**Estimate of infection intensity**. Infection intensity was estimated using two surrogates, eggs per gram of faeces (EPG) and plasma level of circulating anodic antigen (CAA) released from the parasite's gut into the bloodstream. EPG was determined using the Percoll technique with 250 mg of faeces as described in ref. [57]. For each sample, slides were prepared in triplicates of 150 μL, the eggs counted under a microscope and the mean count multiplied by four to give EPG. Circulating anodic antigen (CAA) was measured applying the up-converting reporter particles (UCP) for quantitation of CAA with a lateral flow (LF) test strip as previously described[58,59], using a sandwich with the same mouse monoclonal anti-CAA antibody (clone 147 LUMC, Parasitology) on the LF strip and the UCP. A modified Packard Fluorocount microtiter plate reader was used to measure UCP signals. Standard curves were constructed to determine CAA levels in the rhesus samples and a negative human serum was included. The increasing CAA levels up to Wk10 were analysed by fitting a simple logistic curve to the data using Microsoft Excel (version 15.0.5371.1000) and GraphPad Prism v.8.0 (Supplementary Fig. 1). The levels after Wk10 followed a negative exponential decline and the dataset for each animal between Wk10 and Wk42 was linearised by an Ln-transform, followed by linear regression. This allowed the cure rate to be estimated, where $t1/2 = ln2/\lambda = 0.693/$regression slope. The same formula was used to estimate the rate at which egg excretion decreased between Wk10 and Wk24. An estimate of worm burden in adult worm equivalents was back-calculated from data in the literature[43] on five adult macaques perfused at Wk8, which had means of 424 worms and 5780 pg/ml CAA, respectively, amounting to 13.6 pg CAA/worm.

**Principal components analysis and subject stratification**. We knew from previous work with schistosome infections in rhesus macaques that self-cure has a variable aetiology[18,19]. We, therefore, sought to cluster the ten macaques (excluding Rh1 with lower cercarial exposure and Rh10, withdrawn at Wk10) by performing an unsupervised principal components analysis (PCA) with the prcomp function in R(v 3.6.2)[60]. graphs were plotted with factoextra (v 1.0.7) and FactoMineR (v 2.4) packages in R. The entire set of 15 parameters acquired over 62 weeks was first considered. Nine (CAA, EPG, weight, anti-SCAP, anti-SEA, anti-SWAP, IP-10, SAA and %ATP in cultured schistosomula) were measured at 19 time points (171 events), the other six haematology parameters at 14 time points (84 events), making a total of 255 events; at each event ten datapoints were acquired, one from each subject (2550 datapoints). The least informative events (having three or more datapoints equal to zero) were filtered out, leaving 212 events for analysis (2120 datapoints). The unsupervised PCA (Supplementary Fig. 8a) showed that overall, the 15 parameters were interdependently correlated, the first component explaining 23.4%, the second 16.3% and the third 12.4 % of the variance in the system; the first three PCs explain 52.1% of the variance. The subjects were clustered into three groups, a more homogeneous group comprised Rh2, 3, 4 and 9, a second more dispersed group of Rh5, 6, 8 and 11 and a third separate group of Rh7 and 12. A second unsupervised PCA (Supplementary Fig. 8b) with only five selected parameters (CAA, EPG, anti-SCAP, anti-SEA and anti-SWAP) sharpened the subject clustering into three groups, and increased the variance explained by the first and second components to 32.6 and 17.2%, respectively; with the five selected parameters, the first two PCs (49.8%) seemed to already mimic the separation with three components (52.1%) obtained when including all 15 parameters. This provides the statistical underpinning for the separation of the subjects into Fast, Intermediate and Slow groups in relation to their cure rates (see Results). Thus, we adapted a practice from clinical medicine, namely to analyse the differential responses to therapy in stratified groups within the population[61,62]. Considering self-cure as the 'therapy', we used the level of CAA in the bloodstream, one of the 'response biomarkers' that permitted stratification of the subjects, to calculate the cure rates, which reflect the differential efficacy of 'therapy' between the Fast and Slow responder groups (see Results).

**Kinetics of immune and antibody responses as well as of haematological changes**. We interrogated the profile of specific IgG production over the 62-week time course, using four soluble antigen preparations from the infective cercaria (SCAP), migrating schistosomulum (SSP), mature egg (SEA) and adult worm (SWAP). All the preparations share housekeeping proteins of the cytosol and cytoskeleton, but each also contains stage-specific proteins, largely secretory and related to life cycle functions (Summarised in Supplementary Fig. 7b).

SSP and SWAP were prepared as described in ref. [63]. ELISA for SSP and SWAP were performed as described in ref. [64], with minor modifications: microplates were coated overnight at 4 °C with a concentration of 0.43 μg/mL (100 μL/well) of the different antigens diluted in carbonate buffer (pH 9.6). For IgG ELISA, plasma was diluted 2000-fold. Care was taken to ensure that the ELISA results were directly comparable as possible. All ELISAs were performed on the same day, in parallel and using common buffers and substrate reagents. Times for antibody incubations and substrate development were identical for all plates. Levels of IgG against SCAP and SEA were determined by Antibody–UCP-LF– assay, as described previously[20]. In short: plasma samples (1 μL) were diluted in assay buffer and analysed on two LF strips, one containing a Test line of SCAP and the other a Test line of SEA. A

consecutive flow format was used to detect antibody binding to the SCAP or SEA Test line with UCP reporter particles coated with protein-A[65].

The inflammatory status of the animals was monitored using two plasma markers[66,67], namely serum amyloid A (SAA) and interferon γ-inducible protein 10 (IP-10). SAA is an acute-phase protein of the short pentraxin family, produced in hepatocytes, primarily in response to IL-6 (ref. [68]). IP-10 (CXCL10) is a chemokine produced by several leucocyte types in response to IFNγ stimulation, which in turn binds to the CXCR3 receptor, recruiting activated Th1 lymphocytes to sites of inflammation[69]. The levels of SAA and IP-10 were measured using the UCP-LF platform as previously described[58,59], with two different test strips specific for SAA and IP-10. The antibodies used were: mouse monoclonal for SAA sandwich: on the LF strip mAb anti-SAA1 clone SAA15 [NB100-73077], the UCP conjugate with mAb clone SAA1 [NB100-73071] (both from Novus Biologicals); and mouse monoclonal for IP-10 sandwich: on the LF strip mAb anti-IP-10 clone B-C55 [879.950], the UCP conjugate with mAb clone B-C50 [855.420] (both from Diaclone Research).

Erythrocyte and leucocyte counting, as well as haemoglobin determination, were performed automatically in an ABX Pentra XL 80 instrument (Horiba). Differential leucocyte counts were performed on one hundred cells in a blood smear using an Eclipse E200 microscope (Nikon). Haematocrit was determined with the micro-capillary technique, using a Fanem 3400 centrifuge.

**Recovery and morphological analysis of surviving worms**. Perfusions were performed at Wk62 (Wk20pc). Rhesus macaques were initially sedated with a mixture of ketamine hydrochloride and xylazine for blood and faeces collection, as described above. Heparin (10,000 IU) was then injected via the femoral vein and after 5 min the animals were euthanized with a lethal dose of sodium thiopental (21 mg/kg) and ketamine hydrochloride (36 mg/kg) as previously described[19,70]. RPMI-1640 medium (Gibco, Life technologies) buffered with 10 mM HEPES was infused into the aorta via a peristaltic pump, and the perfusate was collected at the portal vein outlet. The recovered parasites were harvested from the perfusate, washed in PBS, counted per rhesus and fixed in Karnovsky's solution (16% paraformaldehyde, 0.2 M sodium phosphate buffer, 50% glutaraldehyde, 0.2 M cacodylate buffer, pH 7.4) for confocal microscopy. Worms were prepared for confocal microscopy by staining in Langeron's Carmine, dehydration and mounting as previously described[18,71]. Optical slices were obtained on a Zeiss LSM710 confocal microscope, with excitation at 514 nm from a 25-mW argon-ion laser and a 585 nm long-pass emission filter.

**Schistosomula in vitro culture**. Cercariae shed from infected snails were cooled on ice for 30 min and collected by centrifugation. Schistosomula were obtained by mechanical transformation of cercariae, and separation of their bodies as previously described[72]. The newly transformed schistosomula (NTS) were maintained for 72 h, before coculture, in M169 medium (Vitrocell, cat number 00464) supplemented with penicillin/streptomycin, amphotericin, gentamicin (Vitrocell, cat number 00148), 1 μM serotonin, 0.5 μM hypoxanthine, 1 μM hydrocortisone and 0.2 μM triiodothyronine at 37 °C and 5% $CO_2$.

**Schistosomula coculture with plasma and viability evaluation**. For viability evaluation by ATP measurement, NTS were maintained in 96 well plates, 100 schistosomula per well in 100 μl of supplemented M169 medium for 72 h. A 50 μl sample of plasma from each rhesus/time point or 50 μl of serum from each hamster/time point was then added to individual wells (plasma final concentration of 33% v/v). Parasites in medium only were used as controls. After a further 48 h of incubation, the viability of schistosomula was determined by a cytotoxicity assay[73], based on the CellTiter-Glo Luminescent Cell Viability Assay (G7570, Promega), which determines the amount of ATP present in schistosomula as an indication of metabolically active cells. Three biological replicates of schistosomula preparations were analysed for each rhesus/time point and two biological replicates (each containing a pool of sera from three hamsters) of schistosomula preparations were analysed for each hamster/time point. Two technical replicates were assayed for each biological replicate. Schistosomula motility was recorded using an Olympus CKX41 inverted microscope. Schistosomula motility was strongly affected by treatment with rhesus plasma collected later than at Wk12, including at Wk16 (Supplementary Movie 1) or Wk20pc (Supplementary Movie 2) when compared with control schistosomula (Supplementary Movie 3).

Viability was also evaluated by staining for live and dead schistosomula using propidium iodide (PI; Sigma-Aldrich) and fluorescein diacetate (FDA; Life Technologies)[74]. In live cells, FDA is converted into charged fluorescein by esterase activity. NTS were distributed at a density of 400 per well in 24-well plates and incubated for 72 h as described above. They were then incubated with pools of plasma from Fast-responders (Rh2, 3, 4 and 9) or Slow responders (Rh5, 6, 8 and 11) for another 72 h before 2 μg/ml PI plus 0.5 μg/ml FDA were added. Controls with no plasma were assayed in parallel. The schistosomula were immediately observed by light microscopy at 10x magnification using an Olympus CKX41/U-RFLT50 fluorescent inverted microscope, with death recorded by a red fluorescence signal (572 nm emission filter) and live schistosomula by a green signal (492 nm emission filter).

**ChIP-seq**. We assessed whether the action of rhesus plasma on in vitro developing schistosomula might cause a change in their epigenetic programme that could affect their ability to resist immunological pressure, as it was recently shown that histone methylation changes are required for life cycle progression in S. mansoni[75]. We wished to evaluate, and map to the susceptible genomic loci, the possible epigenetic changes at histone H3 trimethylated lysine 4 (H3K4me3) chromatin marks, usually present at the transcription start site (TSS) of genes. To do this we performed chromatin immunoprecipitation followed by high-throughput sequencing (ChIP-Seq) experiments. Thus, 3-day-old schistosomula were cultured for 48 h with a pool of plasma from all rhesus macaques collected at Wk0, Wk8 or Wk10. The parasites were then stabilised using formaldehyde, lysed and sonicated with the Bioruptor Pico (Diagenode) for five cycles (30 s ON and 30 s OFF). Sheared chromatin was collected and used on the IP-STAR Auto ChIPmentation Kit for Histones (Diagenode cat #C01011000) according to the User Guide v 2 01_02_2018 and screen instructions (Diagenode). Ab coating time was changed to 3 h, IP reaction to 13 h, washes to 10 min and tagmentation to 5 min. For each sample, the Ab coating mix was prepared with 4 μl anti-H3K4me3 (Diagenode C15410003; lot A1051D and A1052D). Empty beads served as a negative control. Stripping, end repair and reverse cross-linking were done as indicated in the User Guide. The libraries were quantified using Qubit dsDNA HS Assay Kit (Q33230, Thermo Fischer Scientific) and fragment size was checked on an Agilent 2100 Bioanalyzer with a High Sensitivity DNA Assay. Paired-end sequencing (2 × 75 cycles) was performed on a NextSeq 550 instrument (Illumina, USA).

Quality check of reads was performed using FastQC (v.0.11.7, https://www.bioinformatics.babraham.ac.uk/projects/fastqc/). Fastp[76] (v0.20.0) was used to trim adaptors and reads with low sequencing quality. Reads were mapped using bowtie2 (v.2.2.9)[77] against the S. mansoni genome PRJEA36577 (v7) retrieved from WormBase (schistosoma_mansoni.PRJEA36577.WBPS14.genomic_softmasked.fa) and the overall average mapping rate of ChIP-Seq reads to the genome was 93% (Supplementary Data 10). Default parameters were used to report only the best alignment of each paired-end read. Samtools[78] (v.1.8) and picard-tools MarkDuplicates (v.1.95) (https://broadinstitute.github.io/picard/) were used to filter and remove PCR and optical duplications; filtering resulted in ~58% of mapped reads remaining for further analysis (Supplementary Data 10). Qualimap[79] (v.2.2.1) was used for mapping quality control. After removal of reads mapping to mitochondria, peak calling was performed with MACS2 (v 2.1.1)[80] using the AQUAS pipeline (https://github.com/NHLBI-BCB/TF_chipseq_pipeline). Peak analyses showed that on average ~2500 peaks were identified per sample with an average peak length of 170 bp (Supplementary Data 11), at a significance threshold $p$ value of 0.01. H3K4me3 peaks can be visualised in the genome browser at http://schistosoma.usp.br. H3K4me3 mark peaks were assigned to genes according to the annotation published by our group[81] comprising 16,583 lncRNAs and 14,520 protein-coding genes annotated in the S. mansoni genome (v7). Peaks were assigned only when located within 2 kb upstream from the TSS or along the gene body. Peaks were assigned to three categories: TSS peaks (located within 2 kb up or downstream from gene TSS), Gene Body peaks (peaks located downstream >2 kb from gene TSS) and IGR peaks (intergenic region peaks, not present in TSS or Gene Body peak groups). About 78% of H3K4me3 marks had at least one transcript assigned (Supplementary Data 11). Peaks with differential H3K4me3 mark abundance detected after plasma treatment were identified with DiffBind[82] (2.12.0). Upon treatment of schistosomula with rhesus plasma, dynamic changes of histone marks location with respect to gene position were observed for H3K4me3 marks (chi-square $X = 390.63$, df = 8, $p$ value <2.2E-16). For example, the H3K4me3 Wk8 sample showed the highest number of peaks around TSS ( ± 2 kb), with ~50% of peaks located near the TSS of protein-coding and lncRNA transcripts, and ~32% along the gene body; this pattern was modified in the H3K4me3 Wk10 sample, with 31% of H3K4me3 marks located near the TSS, while 53% were identified along the gene body. Pathways related to the genes affected by plasma treatment were identified using the GO annotation obtained with eggNOG[83] and the BINGO[84] tool to calculate the statistically significant overrepresented Gene Ontologies (GOs) (FDR <1%). In order to avoid redundant GOs the algorithm REVIGO[85] was used with the whole UniProt GO terms as database.

**RNA-seq**. For RNA-Seq assays, 3-day-old schistosomula were cocultured for 72 h with a pool of plasma from all rhesus macaques collected at Wk0, Wk8 or Wk43 (i.e. Wk1pc). Schistosomula not exposed to plasma were used as controls. Three biological replicates from each condition were assayed. Total RNA from schistosomula was extracted using the RNeasy Micro Kit (Qiagen, 74004). RNA samples were quantified using the Qubit RNA HS Assay Kit (Thermo Fisher Scientific, Q32852); purity was evaluated using NanoDrop ND-1000 Spectrophotometer (NanoDrop Technologies) and the integrity was verified using the Agilent RNA 6000 Pico Kit (Agilent Technologies, 5067-1513) in the 2100 Bioanalyzer Instrument (Agilent Technologies). Stranded tagged cDNA libraries were prepared by BGI Genomics using the Strand-Specific Transcriptome Library Construction Protocol (DNBSEQ, SOP-SS-115) and quantification and quality were performed on a 2100 Bioanalyzer (Agilent). Libraries were pooled and sequenced (300 cycles, paired-end sequencing) on an MGISEQ-2000 instrument. An average of 34.1 million paired-reads per sample was obtained (Supplementary Data 12). Quality check of reads and trimming of adaptors and reads with low sequencing quality

were performed as described for ChIP-Seq. RNA-Seq reads were mapped using STAR[86] (v 2.7.3a) against the *S. mansoni* genome PRJEA36577 (v7) retrieved from WormBase (schistosoma_mansoni.PRJEA36577.WBPS14.genomic_softmasked.fa) and the overall average mapping rate was 98% (Supplementary Data 12). Read counting was performed with RSEM[87] (v 1.3.1) using a previously published transcriptome annotation[81]. Read count values (Supplementary Data 6) are shown as log2CPM normalised across all conditions with Trimmed Mean of M-values (TMM) method implemented in the edgeR package within the R platform (v 3.6.2)[60]. A manual search for the genes of the early autophagy pathway[88] was performed using curated sequences for yeast and human on Uniprot (https://www.uniprot.org/) searched by BLASTp against NCBInr (https://blast.ncbi.nlm.nih.gov) and WormBase Parasite (https://parasite.wormbase.org). Genewise dispersion estimation was calculated with edgeR using all 12 samples, and statistical analysis for identification of differentially expressed genes (DEGs) was performed by comparing each cocultured group to control with two different algorithms: (i) limma+voom[89] (v 3.44.3) and (ii) edgeR+svaseq[90,91] (v 3.30.3). Only the genes found differentially expressed in both analyses (FDR $\leq 5 \times 10^{-2}$) were reported. The DEGs are shown with an unsupervised heatmap clustering that was built using the pheatmap (v 1.0.12) function in the R platform (v 3.6.2)[60]. Pathways related to these DEGs were identified using the published egg-NOG.HMM.Description and GO annotation of *S. mansoni* genes[81] obtained with eggNOG[83], and the BINGO[84] tool to calculate the statistically significant over-represented GOs (FDR <5%).

**Reporting Summary**. Further information on research design is available in the Nature Research Reporting Summary linked to this article.

## Data availability

The ChIP-Seq and RNA-Seq data generated in this study have been deposited in the NCBI Sequence Read Archive (SRA) under Accession number PRJNA602708. All other data supporting the findings of this study are available within the article and its Supporting Information files. Source data are provided with this paper as a Source Data file. Source data are provided with this paper.

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

## Acknowledgements
We would like to acknowledge Professor Richard Law, York Centre for Complex Systems Analysis, for advice on the analysis of datasets. We thank the technical and veterinary staff from the Central Animal Facility from Butantan Institute, São Paulo, Brazil, for proper and careful handling of the animals throughout the study. Epigenetics experiments were done on the Environmental Epigenetics facility of IHPE with the support of LabEx CeMEB, an ANR 'Investissements d'avenir' programme (ANR-10-LABX-04-01). We thank Jean-François Allienne and the 'Bio-Environnement' platform at the University of Perpignan Via Domitia for support in NGS library preparation and sequencing. We gratefully acknowledge the stimulating discussions throughout the entire project of Dr. Ricardo DeMarco, Universidade de São Paulo at São Carlos, in memoriam. This work was supported by grants from Fundação de Amparo à Pesquisa do Estado de São Paulo (FAPESP) (Grant numbers 15/06366-2, 18/15049-9 and 18/23693-5 to S.V.-A). D.W.S. received a fellowship from Coordenação de Aperfeiçoamento de Pessoal de Nível Superior (CAPES), Brazil – Finance Code 001 and subsequently from FAPESP (19/09404-3); A.S.A.P. is a fellow of FAPESP (16/10046-6); J.V.M.M. is a fellow of FAPESP (18/18117-5); C.G. received a grant from EPICURE (projet de recherche conjoints CNRS Brésil FAPESP – 2018) and the support of LabEx CeMEB, an ANR 'Investissements d'avenir' programme (ANR-10-LABX-04-01). SVA laboratory was also supported by institutional funds from Fundação Butantan; S.V.-A. received an established investigator

fellowship award from CNPq, Brazil. The funders had no role in study design, data collection and analysis, decision to publish or preparation of the manuscript.

## Author contributions

M.S.A., R.A.W. and S.V.-A. conceived the experiments. M.S.A., V.G.M.M., R.C.A., C.G., R.A.W. and S.V-A. designed the experiments. M.S.A., D.W.S, A.S.A.P, J.V.M.M, P.A.M., R.d.P.F., E.M.T.K.F., C.J.d.D., S.d.O.C. and R.d.C.A. conducted experiments. M.S.A, A.C.T., R.A.W. and S.V-A. performed data analysis. J.K., P.L.A.M.C., G.J.v.D., E.N., V.G.d.M.M., C.G., R.A.W. and S.V-A. provided resources. M.S.A., R.A.W. and S.V.-A. wrote the manuscript. D.W.S., A.C.T., P.A.M., J.K., P.L.A.M.C., E.N., S.d.O.C., V.G.d.M.M., R.d.C.A. and C.G. commented on the manuscript. R.A.W. and S.V.-A. revised and edited the manuscript. S.V.-A. coordinated the study. All authors approved the final version of the article.

## Competing interests

The authors declare no competing interests.
