## [Peer Review File · Nature Communications]

Peer review initial comments –

Reviewer #1 (Remarks to the Author):

This is a well-written research article that focuses on contributing to our knowledge of the mechanism(s) involved in the rhesus macaques self-curing process from schistosome infections. Overall an excellent piece of work!

Please see below for my comments on the manuscript.

1. In the Methods, the authors estimated infection intensity by using 2 surrogates, fecal egg per gram and plasma CAA. While CAA has been demonstrated to have high levels of specificity and sensitivity, fecal egg microscopy, on the contrary, has very low sensitivity (less than 50%). Is there any particular reason why a more sensitive method, like fecal egg PCR, was not used instead? The fecal PCR would have, been more informative particularly during the self-curing phase with just few worms remaining as these infected macaques could still be egg-positive even though the eggs could no longer be detected by microscopy.

2. In the Results section (lines 193-198), comparing the length of so few male (n=9) and female worms (n=2) recovered from the macaques to those from infected mice does not provide sufficient power for statistical analysis. I strongly suggest that the authors tone down any inference drawn from the worm length data. Also, there is nowhere in the Methods where the authors described infecting mice with *Schistosoma mansoni*. Did the authors intend to refer to hamsters instead?

3. Is there any particular reason why only female rhesus macaques (and no males) were used in this study? Would the results have been different with both sexes used?

4. In the Results section, the authors stated that there was significant change in H3K4me3 abundance at 76 different genomic regions in 3-day old schistosomula co-cultured with rhesus plasma collected at Wk10 when compared to Wk0 (lines 322-325). Did the authors perform any other experiments, such as qPCR to verify and/or validate the expression of some of the genes identified as being divergently transcribed? I believe carrying out qPCR (at the least) to validate the expression of these genes will strengthen the claims on the epigenetic reprogramming of schistosomula.

5. In the Discussion, the authors suggested that the regulation of genes that promote autophagy may have contributed to the parasite starvation caused by immune pressure, leading to parasite death and to rhesus self-cure (lines 402-404). This hypothesis could have been easily verified by the authors by analyzing the expression profiles of these autophagy-related genes in the 3-day-old schistosomula co-cultured with Wk10 rhesus plasma versus Wk0 plasma control.

Reviewer #3 (Remarks to the Author):

A single drug of only modest efficacy is available for treating schistosomiasis. Since this disease affects 200 million people around the world, it is critical to develop new treatment modalities. One of the most attractive options is the development of an efficacious vaccine against the parasite that would alleviate the need of annual drug administration to keep the disease at bay. Unfortunately, an effective vaccine target has remained elusive. Thus, understanding the interaction between the parasite and the immune system is critical. Here, Amaral et al. experimentally describe the process of schistosome elimination, or "self-cure", in Rhesus macaques. Although, this is not a new observation, this study brings a number of newer tools to the table to examine the process. While this is a totally valid and worthwhile endeavor, the analysis of the data are superficial, there are no functional studies characterizing the self-cure process and thus the study brings no molecular insights readily actionable against the disease or to understanding basic parasite or mammal biology. Although CHIP-seq studies were performed, it is unclear what they

add to the study. In summary, this is an interesting and worthwhile study that will be of interest to researchers in the schistosome community.

REVIEWER COMMENTS

Reviewer #1 (Remarks to the Author):

This is a well-written research article that focuses on contributing to our knowledge of the mechanism(s) involved in the rhesus macaques self-curing process from schistosome infections. Overall an excellent piece of work!

Authors response:

We thank Reviewer 1 for the above general comments. Please see below for our response to the specific comments.

Please see below for my comments on the manuscript.

1. In the Methods, the authors estimated infection intensity by using 2 surrogates, fecal egg per gram and plasma CAA. While CAA has been demonstrated to have high levels of specificity and sensitivity, fecal egg microscopy, on the contrary, has very low sensitivity (less than 50%). Is there any particular reason why a more sensitive method, like fecal egg PCR, was not used instead? The fecal PCR would have, been more informative particularly during the self-curing phase with just few worms remaining as these infected macaques could still be egg-positive even though the eggs could no longer be detected by microscopy.

Authors response:

Reviewer 1 notes that we used fecal egg microscopy as one of the surrogates for infection intensity and comments that it has a very low sensitivity (less than 50%). Is there any particular reason why a more sensitive method, like fecal egg PCR, was not used instead?

The first point to clarify is that we did not use the conventional Kato/Katz faecal smear technique, which does indeed lack sensitivity. We used the Percoll density gradient technique of Eberl et al (2002), specifically developed to estimate infection intensity in primate experiments. When validated against human infections in Egypt, the technique had a sensitivity of 71% on duplicate Percoll estimates versus 52% for triplicate Kato/Katz smears. The Percoll sensitivity rose to 100% on a second visit while that of the Kato/Katz smear remained ~50%. The Percoll method is unsuitable for large-scale field surveys since it requires an accurate balance, a laboratory centrifuge, and fresh faeces (the eggs must be live), but that was not an issue in our laboratory-based rhesus macaque experiment.

The second point is why did we not use PCR on faecal samples to estimate infection intensity and the answer is simple – a lack of quantitation. The PCR approach was first described in 2003 (Pontes et al., *Am. J. Trop. Med. Hyg.*, 68(6), 652–656). Applied as an alternative diagnostic tool to the Kato/Katz smear it proved more sensitive but only in a positive/negative sense, not quantitatively. Subsequent improvements have introduced a semi-quantitative element by using PCR cycle number as the indicator. However, the cycle number at which the reaction became positive was “calibrated” against comparable Kato/Katz smears on the same samples, being divided into high, medium, low, and negative infection categories (Meurs et al., 2015, *PLoS Negl Trop Dis* 9(7): e0003959.) There was a correlation of >0.8 between the two methods and the authors concluded 13–15% more infections were detected by PCR when compared to microscopy on a single stool smear sample. Based on the Meurs approach, the PCR method has been deployed in field studies (e.g. Al Shehri et al., 2018, *Parasitology* 145, 1715–1722.) with infection intensity classified according to cycle time (Ct) values: negative (Ct > 45), light positive (35 > Ct ≤ 45), medium

positive ($25 > Ct \leq 35$), and heavy positive ($Ct \leq 25$). That is as far as quantitation of the PCR approach has progressed in any study.

We would like to argue that the Percoll method, with demonstrable greater sensitivity than the Kato/Katz smear, is at least as sensitive as PCR on faecal samples, and most importantly the eggs per gram of faeces obtained with the Percoll method provides a numerical value for the surrogate estimate of infection intensity.

2. In the Results section (lines 193-198), comparing the length of so few male (n=9) and female worms (n=2) recovered from the macaques to those from infected mice does not provide sufficient power for statistical analysis. I strongly suggest that the authors tone down any inference drawn from the worm length data. Also, there is nowhere in the Methods where the authors described infecting mice with *Schistosoma mansoni*. Did the authors intend to refer to hamsters instead?

Authors response:

Following the reviewer's suggestion, we have now removed the statistical analysis related to the sizes of the worms recovered from macaques compared to worms recovered from mice, and we kept only the description of the length and width of the worms without statistics (Lines 198-200).

Apologies for the omission in the Methods of the source of worms recovered from mice. We have now added the corresponding description to the Methods (Lines 601-603).

3. Is there any particular reason why only female rhesus macaques (and no males) were used in this study? Would the results have been different with both sexes used?

Authors response:

We used only females in this study because of their availability in the Butantan colony. Rhesus macaque colonies are a scarce resource worldwide, and they are very socializing animals that live in a family with a dominant male. The colony has been managed by the Butantan Institute veterinarians since 1929, and they keep only a few dominant males among tens of females, to avoid males fighting.

To the best of our knowledge, no previous studies have compared male versus female rhesus macaques infected with *S. mansoni*.

4. In the Results section, the authors stated that there was significant change in H3K4me3 abundance at 76 different genomic regions in 3-day old schistosomula co-cultured with rhesus plasma collected at Wk10 when compared to Wk0 (lines 322-325). Did the authors perform any other experiments, such as qPCR to verify and/or validate the expression of some of the genes identified as being divergently transcribed? I believe carrying out qPCR (at the least) to validate the expression of these genes will strengthen the claims on the epigenetic reprogramming of schistosomula.

Authors response:

Following the reviewer's suggestion that the expression profiles of genes with significant change in H3K4me3 abundance should be evaluated, we have performed a transcriptome-wide RNA-Seq experiment. This permitted us to evaluate the impact of rhesus plasma collected at Wk0, Wk8 and Wk1pc on the expression levels of all genes in *in vitro* cultured schistosomula. Wk8 was chosen because it is the earliest time point at which a decrease in viability of schistosomula was observed *in vitro* but parasite death was not yet evident *in vivo*, as shown by a continued increase in CAA levels. Wk1pc was chosen as the earliest time

post-challenge where an augmented protective antibody attack on schistosomula was evident. The *in vitro* co-culture of 3-day-old schistosomula with plasma should recapitulate the *in vivo* condition and highlight critical pathways related to parasite commitment before death.

We found in the RNA-Seq a total of 79 differentially expressed genes (DEGs) when schistosomula co-cultured with Wk8 plasma were compared with control schistosomula; no GOs were enriched among these. In marked contrast, a total of 293 DEGs were found when schistosomula co-cultured with Wk1pc plasma compared with controls, 50 of them shared with week 8 plasma cultures. Remarkably, only three GO categories were enriched among the 293 DEGs, all of them representing GO cellular components related to late autophagic processes: “lytic vacuole”, “lysosome” and “vacuole”. Nine of the ten genes in these GO categories were down-regulated suggesting that the immune pressure disrupts expression of genes involved in autophagy, ultimately leading to parasite death and to self-cure. These new Results were added to the text (lines 343 to 360).

5. In the Discussion, the authors suggested that the regulation of genes that promote autophagy may have contributed to the parasite starvation caused by immune pressure, leading to parasite death and to rhesus self-cure (lines 402-404). This hypothesis could have been easily verified by the authors by analyzing the expression profiles of these autophagy-related genes in the 3-day-old schistosomula co-cultured with Wk10 rhesus plasma versus Wk0 plasma control.

Authors response:

See the interpretation of RNA-Seq data above. In addition, since schistosome autophagy is a very under-researched topic, we performed a manual curation of prospective early-stage autophagy genes in *S. mansoni*, identifying 15 that were present in our RNA-Seq dataset. However, none was differentially expressed in response to plasma treatment. The nine downregulated genes in the three enriched GO categories are all cellular components involved in late stages of autophagy, suggesting a dysfunction of the processes that remodel the schistosomulum body after infection. We also observed the 12-fold upregulation of a protease inhibitor that could potentially interfere with the elongation of lung schistosomula, prior to their migration through capillary beds, *in vivo*. We therefore suggest that autophagy dysfunction may contribute to the plasma-mediated parasite death observed *in vitro* and to the starvation caused by immune pressure *in vivo*, leading to parasite death and to rhesus self-cure. The Discussion has been modified (lines 422 to 445).

Reviewer #3 (Remarks to the Author):

A single drug of only modest efficacy is available for treating schistosomiasis. Since this disease affects 200 million people around the world, it is critical to develop new treatment modalities. One of the most attractive options is the development of an efficacious vaccine against the parasite that would alleviate the need of annual drug administration to keep the disease at bay. Unfortunately, an effective vaccine target has remained elusive. Thus, understand the interaction between the parasite and the immune system is critical. Here, Amaral et al. experimentally describe the process of schistosome elimination, or “self-cure”, in Rhesus macaques. Although, this is not a new observation, this study brings a number of newer tools to the table to examine the process. While this is a totally valid and worthwhile

endeavor, the analysis of the data are superficial, there are no functional studies characterizing the self-cure process and thus the study brings no molecular insights readily actionable against the disease or to understanding basic parasite or mammal biology. Although CHIP-seq studies were performed, it is unclear what they add to the study. In summary, this is an interesting and worthwhile study that will be of interest to researchers in the schistosome community.

Reviewer 3 notes that while our study brings a number of newer tools to the table to examine the self-cure process, the analyses of the data are superficial.

Authors response:

As mentioned in the Acknowledgements, given the extreme variability in response and outcome between the individual macaques, we took the advice of Prof Richard Law of the York Centre for Complex Systems Analysis. He suggested segmental analysis of the data into three phases: establishment, self-cure and post challenge, and then use of a curve fitting approach to describe the kinetics of each process. The parameter values obtained allowed objective stratification of the individual macaques, along the lines of clinical practice with human patients. Due to length restrictions, we did not describe the principal component analysis (PCA) that underpinned the subject stratification. The unsupervised PCA with data from all 15 parameters evaluated over the 62 weeks of experimentation, acquired from 10 rhesus macaques has been added to the Methods (lines 625 to 654). Overall, the 15 parameters were interdependently correlated, and the subjects clustered into Fast, Intermediate and Slow responder groups, in line with the respective cure rates. The first three components of the PCA accounted for 52.1% of the variance in the system. A second unsupervised PCA with only five selected parameters, namely CAA, EPG, anti-SCAP, anti-SEA, and anti-SWAP, reinforced the subject clustering into the three groups, with the first and second components accounting for 49.8% of the variance. See Results (lines 115 to 119).

Reviewer 3 also considers that there are no functional studies characterizing the self-cure process and thus the study brings no molecular insights.

Authors response:

Identification of the antigenic targets and detailed mechanisms of the self-cure process is indeed a crucial goal. However, the self-cure model has only recently been revived (Ref 18, 2008; Ref 19, 2015) after decades of inaction, so there are few pointers in the literature.

For functional studies we developed the *in vitro* cultivation system for schistosomula, with macaque plasma added between Days 3 and 6. This parallels the intravascular migrating “lung” stage, which may be the earliest target of the self-cure response. Note that our procedure is not the same as the ADCC assay with newly transformed 3hr schistosomula, frequently described in the literature. We first used the assay with plasma from the complete time course, using %ATP as the readout, and demonstrated its increasing potency to inhibit parasite metabolism. The impact became appreciable between weeks 8 and 16 before reaching a plateau, and then increased again immediately after challenge.

We also used the experimental system to look for epigenetic marks that might be changed in Day 3-6 schistosomula co-cultured with rhesus plasma from weeks 0, 8 and 10. A number of gene loci had lost the H3K4me3 histone mark at their transcription start sites upon exposure to Wk10 plasma.

In the revised manuscript we now report the use of the system in a third assay, a genome wide RNA-Seq experiment to look for changes in gene expression elicited by exposure of Day 3-6 larvae to plasma from Wk0, Wk8 and Wk1pc. As might be anticipated the Wk1pc plasma, very active in the %ATP assay, altered the expression level of 293 genes, far more than the Wk8 plasma.

This brings us to the question of molecular insights.

Authors response:

Our aim in the use of Chip-Seq was to identify genetic loci where epigenetic chromatin marks were affected by exposure of schistosomula to self-cure plasma. GO analysis identified “negative regulation of autophagy” as one of the Biological Processes enriched with genes whose epigenetic marks were affected. Remarkably, and independently, when we examined the 293 genes affected by incubation of schistosomula with Wk1pc plasma, only three GO categories were enriched, all of them representing GO cellular components related to late autophagic processes: “lytic vacuole”, “lysosome” and “vacuole”. Nine out of the ten genes were downregulated, suggesting inhibition of normal autophagy processes by the immune pressure. These are believed important in remodeling of the schistosomulum body after infection and before migration through the vasculature.

Armed with this information we identified, by manual curation, 19 Smp genes that are orthologs of 15 genes of the early autophagy pathway. Their expression levels were not affected by the plasma treatments, which were probably applied too late to catch the inception of autophagy within 3hrs of transformation. However, the protein-coding gene most highly upregulated (11.4-fold) among the 293 genes affected by Wk1pc plasma was a serpin peptidase inhibitor. At Day 6 the lung stage schistosomulum is undergoing further tissue reorganisation, involving dissolution of the sub-tegumental fibrous interstitial layer to permit body elongation for migration through pulmonary capillary beds. Upregulation of the inhibitor could thus impair the activity of the proteases removing the fibrous layer and so impede migration *in vivo*. These new Results were added to the text (lines 343 to 360).

The much better researched radiation-attenuated cercarial vaccine model (>500 papers in PubMed) has recently been subjected to both systems biology analysis to identify putative mechanism of protection, and extensive epitope mapping to identify vaccine targets (Farias et al., 2021a & 2021b, *Front Immunol* **11**, 624191 and 624613). This project took several years, performed by Instituto Butantan researchers and partners. We similarly envisage that the novel information on the self-cure rhesus model provided here will spark the interest of the community and open new avenues for investigation that will be fully explored in years to come. We concur with Reviewer 3 that our study will be of interest to researchers in the schistosome community, and we thank him/her for the criticisms and comments.

Peer review further comments on revision –

Reviewer #1 (Remarks to the Author):

Authors have adequately addressed my comments and have made appropriate adjustments in the revised manuscript. This revised version of manuscript is acceptable to me for publication

Reviewer #3 (Remarks to the Author):

I have already conveyed my opinion regarding the potential general and field-specific interest of this manuscript.